# DENOISING DIFFUSION SAMPLERS

**Francisco Vargas**[1]*, **Will Grathwohl**[2] **& Arnaud Doucet**[2]
[1] University of Cambridge, [2] DeepMind

## ABSTRACT

Denoising diffusion models are a popular class of generative models providing state-of-the-art results in many domains. One adds gradually noise to data using a diffusion to transform the data distribution into a Gaussian distribution. Samples from the generative model are then obtained by simulating an approximation of the time-reversal of this diffusion initialized by Gaussian samples. Practically, the intractable score terms appearing in the time-reversed process are approximated using score matching techniques. We explore here a similar idea to sample approximately from unnormalized probability density functions and estimate their normalizing constants. We consider a process where the target density diffuses towards a Gaussian. Denoising Diffusion Samplers (DDS) are obtained by approximating the corresponding time-reversal. While score matching is not applicable in this context, we can leverage many of the ideas introduced in generative modeling for Monte Carlo sampling. Existing theoretical results from denoising diffusion models also provide theoretical guarantees for DDS. We discuss the connections between DDS, optimal control and Schrödinger bridges and finally demonstrate DDS experimentally on a variety of challenging sampling tasks.

## 1 INTRODUCTION

Let $\pi$ be a probability density on $\mathbb{R}^d$ of the form

$$\pi(x) = \frac{\gamma(x)}{Z}, \qquad Z = \int_{\mathbb{R}^d} \gamma(x) \mathrm{d}x, \tag{1}$$

where $\gamma : \mathbb{R}^d \to \mathbb{R}^+$ can be evaluated pointwise but the normalizing constant $Z$ is intractable. We are here interested in both estimating $Z$ and obtaining approximate samples from $\pi$.

A large variety of Monte Carlo techniques has been developed to address this problem. In particular Annealed Importance Sampling (AIS) (Neal, 2001) and its Sequential Monte Carlo (SMC) extensions (Del Moral et al., 2006) are often regarded as the gold standard to compute normalizing constants. Variational techniques are a popular alternative to Markov Chain Monte Carlo (MCMC) and SMC where one considers a flexible family of easy-to-sample distributions $q^\theta$ whose parameters are optimized by minimizing a suitable metric, typically the reverse Kullback–Leibler discrepancy $\mathrm{KL}(q^\theta || \pi)$. Typical choices for $q^\theta$ include mean-field approximation (Wainwright & Jordan, 2008) or normalizing flows (Papamakarios et al., 2021). To be able to model complex variational distributions, it is often useful to model $q^\theta(x)$ as the marginal of an auxiliary extended distribution; i.e. $q^\theta(x) = \int q^\theta(x, u) \mathrm{d}u$. As this marginal is typically intractable, $\theta$ is then learned by minimizing a discrepancy measure between $q^\theta(x, u)$ and an extended target $p^\theta(x, u) = \pi(x)p^\theta(u|x)$ where $p^\theta(u|x)$ is an auxiliary conditional distribution (Agakov & Barber, 2004).

Over recent years, Monte Carlo techniques have also been fruitfully combined to variational techniques. For example, AIS can be thought of a procedure where $q^\theta(x, u)$ is the joint distribution of a Markov chain defined by a sequence of MCMC kernels whose final state is $x$ while $p^\theta(x, u)$ is the corresponding AIS extended target (Neal, 2001). The parameters $\theta$ of these kernels can then be optimized by minimizing $\mathrm{KL}(q^\theta || p^\theta)$ using stochastic gradient descent (Wu et al., 2020; Geffner & Domke, 2021; Thin et al., 2021; Zhang et al., 2021; Doucet et al., 2022; Geffner & Domke, 2022).

Instead of following an AIS-type approach to define a flexible variational family, we follow here an approach inspired by Denoising Diffusion Probabilistic Models (DDPM), a powerful class of

---

*Work done while at DeepMind

generative models (Sohl-Dickstein et al., 2015; Ho et al., 2020; Song et al., 2021c). In this context, one adds noise progressively to data using a diffusion to transform the complex data distribution into a Gaussian distribution. The time-reversal of this diffusion can then be used to transform a Gaussian sample into a sample from the target. While superficially similar to Langevin dynamics, this process mixes fast even in high dimensions as it inherits the mixing properties of the forward diffusion (De Bortoli et al., 2021, Theorem 1). However, as the time-reversal involves the derivatives of the logarithms of the intractable marginal densities of the forward diffusion, these so-called scores are practically approximated using score matching techniques. If the score estimation error is small, the approximate time-reversal still enjoys remarkable theoretical properties (De Bortoli, 2022; Chen et al., 2022; Lee et al., 2022).

These results motivate us to introduce Denoising Diffusion Samplers (DDS). Like DDPM, we consider a forward diffusion which progressively transforms the target $\pi$ into a Gaussian distribution. This defines an extended target distribution $p(x, u) = \pi(x)p(u|x)$. DDS are obtained by approximating the time-reversal of this diffusion using a process of distribution $q^\theta(x, u)$. What distinguishes DDS from DDPM is that we cannot simulate sample paths from the diffusion we want to time-reverse, as we cannot sample its initial state $x$ from $\pi$. Hence score matching ideas cannot be used to approximate the score terms.

We focus on minimizing $\mathrm{KL}(q^\theta||p)$, equivalently maximizing an Evidence Lower Bound (ELBO), as in variational inference. We leverage a representation of this KL discrepancy based on the introduction of a suitable auxiliary reference process that provides low variance estimate of this objective and its gradient. We can exploit the many similarities between DDS and DDPM to leverage some of the ideas developed in generative modeling for Monte Carlo sampling. This includes using the probability flow ordinary differential equation (ODE) (Song et al., 2021c) to derive novel normalizing flows and the use of underdamped Langevin diffusions as a forward noising diffusion (Dockhorn et al., 2022). The implementation of these samplers requires designing numerical integrators for the resulting stochastic differential equations (SDE) and ODE. However, simple integrators such as the standard Euler–Maruyama scheme do not yield a valid ELBO in discrete-time. So as to guarantee one obtains a valid ELBO, DDS relies instead on an integrator for an auxiliary stationary reference process which preserves its invariant distribution as well as an integrator for the approximate time-reversal inducing a distribution absolutely continuous w.r.t. the distribution of the discretized reference process. Finally we compare experimentally DDS to AIS, SMC and other state-of-the-art Monte Carlo methods on a variety of sampling tasks.

## 2 DENOISING DIFFUSION SAMPLERS: CONTINUOUS TIME

We start here by formulating DDS in continuous-time to gain insight on the structure of the time-reversal we want to approximate. We introduce $\mathcal{C} = C([0, T], \mathbb{R}^d)$ the space of continuous functions from $[0, T]$ to $\mathbb{R}^d$ and $\mathcal{B}(\mathcal{C})$ the Borel sets on $\mathcal{C}$. We will consider in this section path measures which are probability measures on $(\mathcal{C}, \mathcal{B}(\mathcal{C}))$, see Léonard (2014a) for a formal definition. Numerical integrators are discussed in the following section.

### 2.1 FORWARD DIFFUSION AND ITS TIME-REVERSAL

Consider the forward noising diffusion given by an Ornstein–Uhlenbeck (OU) process[1]

$$\mathrm{d}x_t = -\beta_t x_t \mathrm{d}t + \sigma\sqrt{2\beta_t}\mathrm{d}B_t, \qquad x_0 \sim \pi, \tag{2}$$

where $(B_t)_{t\in[0,T]}$ is a $d$-dimension Brownian motion and $t \to \beta_t$ is a non-decreasing positive function. This diffusion induces the path measure $\mathcal{P}$ on the time interval $[0, T]$ and the marginal density of $x_t$ is denoted $p_t$. The transition density of this diffusion is given by $p_{t|0}(x_t|x_0) = \mathcal{N}(x_t; \sqrt{1-\lambda_t}x_0, \sigma^2\lambda_t I)$ where $\lambda_t = 1 - \exp(-2\int_0^t \beta_s \mathrm{d}s)$.

From now on, we will always consider a scenario where $\int_0^T \beta_s \mathrm{d}s \gg 1$ so that $p_T(x_T) \approx \mathcal{N}(x_T; 0, \sigma^2 I)$. We can thus think of (2) as transporting approximately the target density $\pi$ to this Gaussian density.

---

[1]This is referred to as a Variance Preserving diffusion by Song et al. (2021c).

From (Haussmann & Pardoux, 1986), its time-reversal $(y_t)_{t\in[0,T]} = (x_{T-t})_{t\in[0,T]}$, where equality is here in distribution, satisfies

$$\mathrm{d}y_t = \beta_{T-t}\{y_t + 2\sigma^2\nabla\ln p_{T-t}(y_t)\}\mathrm{d}t + \sigma\sqrt{2\beta_{T-t}}\mathrm{d}W_t, \qquad y_0 \sim p_T, \tag{3}$$

where $(W_t)_{t\in[0,T]}$ is another $d$-dimensional Brownian motion. By definition this time-reversal starts from $y_0 \sim p_T(y_0) \approx \mathcal{N}(y_0; 0, \sigma^2 I)$ and is such that $y_T \sim \pi$. This suggests that if we could approximately simulate the diffusion (3), then we would obtain approximate samples from $\pi$.

However, putting this idea in practice requires being able to approximate the intractable scores $(\nabla\ln p_t(x))_{t\in[0,T]}$. To achieve this, DDPM rely on score matching techniques (Hyvärinen, 2005; Vincent, 2011) as it is easy to sample from (2). This is impossible in our scenario as sampling from (2) requires sampling $x_0 \sim \pi$ which is impossible by assumption.

## 2.2 REFERENCE DIFFUSION AND VALUE FUNCTION

In our context, it is useful to introduce a *reference* process defined by the diffusion following (2) but initialized at $p_0^{\mathrm{ref}}(x_0) = \mathcal{N}(x_0; 0, \sigma^2 I)$ rather than $\pi(x_0)$ thus ensuring that the marginals of the resulting path measure $\mathcal{P}^{\mathrm{ref}}$ all satisfy $p_t^{\mathrm{ref}}(x_t) = \mathcal{N}(x_t; 0, \sigma^2 I)$. From the chain rule for KL for path measures (see e.g. (Léonard, 2014b, Theorem 2.4) and (De Bortoli et al., 2021, Proposition 24)), one has $\mathrm{KL}(\mathcal{Q}||\mathcal{P}^{\mathrm{ref}}) = \mathrm{KL}(q_0||p_0^{\mathrm{ref}}) + \mathbb{E}_{x_0\sim q_0}[\mathrm{KL}(\mathcal{Q}(\cdot|x_0)||\mathcal{P}^{\mathrm{ref}}(\cdot|x_0))]$. Thus $\mathcal{P}$ can be identified as the path measure minimizing the KL discrepancy w.r.t. $\mathcal{P}^{\mathrm{ref}}$ over the set of path measures $\mathcal{Q}$ with marginal $q_0(x_0) = \pi(x_0)$ at time $t = 0$, i.e. $\mathcal{P} = \arg\min_{\mathcal{Q}}\{\mathrm{KL}(\mathcal{Q}||\mathcal{P}^{\mathrm{ref}}) : q_0 = \pi\}$.

A time-reversal representation of $\mathcal{P}^{\mathrm{ref}}$ is given by $(y_t)_{t\in[0,T]} = (x_{T-t})_{t\in[0,T]}$ satisfying

$$\mathrm{d}y_t = -\beta_{T-t}y_t\mathrm{d}t + \sigma\sqrt{2\beta_{T-t}}\mathrm{d}W_t, \qquad y_0 \sim p_0^{\mathrm{ref}}. \tag{4}$$

As $\beta_{T-t}y_t + 2\sigma^2\nabla\ln p_{T-t}^{\mathrm{ref}}(y_t) = -\beta_{T-t}y_t$, we can rewrite the time-reversal (3) of $\mathcal{P}$ as

$$\mathrm{d}y_t = -\beta_{T-t}\{y_t - 2\sigma^2\nabla\ln\phi_{T-t}(y_t)\}\mathrm{d}t + \sigma\sqrt{2\beta_{T-t}}\mathrm{d}W_t, \qquad y_0 \sim p_T, \tag{5}$$

where $\phi_t(x) = p_t(x)/p_t^{\mathrm{ref}}(x)$ is a so-called value function which can be shown to satisfy a Kolmogorov equation such that $\phi_t(x_t) = \mathbb{E}_{\mathcal{P}^{\mathrm{ref}}}[\phi_0(x_0)|x_t]$, the expectation being w.r.t. $\mathcal{P}^{\mathrm{ref}}$.

## 2.3 LEARNING THE TIME-REVERSAL

To approximate the time-reversal (3) of $\mathcal{P}$, consider a path measure $\mathcal{Q}^\theta$ whose time-reversal is induced by

$$\mathrm{d}y_t = \beta_{T-t}\{y_t + 2\sigma^2 s_\theta(T-t, y_t)\}\mathrm{d}t + \sigma\sqrt{2\beta_{T-t}}\mathrm{d}W_t, \qquad y_0 \sim \mathcal{N}(0, \sigma^2 I), \tag{6}$$

so that $y_t \sim q_{T-t}^\theta$. To obtain $s_\theta(t, x) \approx \nabla\ln p_t(x)$, we parameterize $s_\theta(t, x)$ by a neural network whose parameters are obtained by minimizing

$$\mathrm{KL}(\mathcal{Q}^\theta||\mathcal{P}) = \mathrm{KL}(\mathcal{N}(0, \sigma^2 I)||p_T) + \mathbb{E}_{y_0\sim\mathcal{N}(0,\sigma^2 I)}[\mathrm{KL}(\mathcal{Q}^\theta(\cdot|y_0)||\mathcal{P}(\cdot|y_0))] \tag{7}$$

$$= \mathrm{KL}(\mathcal{N}(0, \sigma^2 I)||p_T) + \sigma^2\mathbb{E}_{\mathcal{Q}^\theta}\left[\int_0^T \beta_{T-t}||s_\theta(T-t, y_t) - \nabla\ln p_{T-t}(y_t)||^2\mathrm{d}t\right],$$

where we have used the chain rule for KL then Girsanov's theorem (see e.g. (Klebaner, 2012)). This expression of the KL is reminiscent of the expression obtained in (Song et al., 2021b, Theorem 1) in the context of DDPM. However, the expectation appearing in (7) is here w.r.t. $\mathcal{Q}^\theta$ and not w.r.t. $\mathcal{P}$ and we cannot get rid of the intractable scores $(\nabla\ln p_t(x))_{t\in[0,T]}$ using score matching ideas.

Instead, taking inspiration from (5), we reparameterize the time-reversal of $\mathcal{Q}^\theta$ using

$$\mathrm{d}y_t = -\beta_{T-t}\{y_t - 2\sigma^2 f_\theta(T-t, y_t)\}\mathrm{d}t + \sigma\sqrt{2\beta_{T-t}}\mathrm{d}W_t, \qquad y_0 \sim \mathcal{N}(0, \sigma^2 I), \tag{8}$$

i.e. $f_\theta(t, x) = s_\theta(t, x) - \nabla\ln p_t^{\mathrm{ref}}(x) = s_\theta(t, x) + x/\sigma^2$. This reparameterization allows us to express $\mathrm{KL}(\mathcal{Q}^\theta||\mathcal{P})$ in a compact form.

**Proposition 1.** *The Radon–Nikodym derivative $\frac{\mathrm{d}\mathcal{Q}^\theta}{\mathrm{d}\mathcal{P}^{\mathrm{ref}}}(y_{[0,T]})$ satisfies under $\mathcal{Q}^\theta$*

$$\ln\left(\frac{\mathrm{d}\mathcal{Q}^\theta}{\mathrm{d}\mathcal{P}^{\mathrm{ref}}}\right) = \sigma^2\int_0^T \beta_{T-t}||f_\theta(T-t, y_t)||^2\mathrm{d}t + \sigma\int_0^T \sqrt{2\beta_{T-t}}f_\theta(T-t, y_t)^\top\mathrm{d}W_t. \tag{9}$$

*From the identity* $\mathrm{KL}(\mathcal{Q}^\theta || \mathcal{P}) = \mathrm{KL}(\mathcal{Q}^\theta || \mathcal{P}^{\mathrm{ref}}) + \mathbb{E}_{y_T \sim q_0^\theta}[\ln\left(\frac{p_0^{\mathrm{ref}}(y_T)}{p_0(y_T)}\right)]$, *it follows that*

$$\mathrm{KL}(\mathcal{Q}^\theta || \mathcal{P}) = \mathbb{E}_{\mathcal{Q}^\theta}\left[\sigma^2 \int_0^T \beta_{T-t} ||f_\theta(T-t, y_t)||^2 \mathrm{d}t + \ln\left(\frac{\mathcal{N}(y_T; 0, \sigma^2 I)}{\pi(y_T)}\right)\right]. \tag{10}$$

For $\theta$ minimizing (10), approximate samples from $\pi$ can be obtained by simulating (8) and returning $y_T \sim q_0^\theta$. We can obtain an unbiased estimate of $Z$ via the following importance sampling identity

$$\hat{Z} = \frac{\gamma(y_T)}{\mathcal{N}(y_T; 0, \sigma^2 I)} \frac{\mathrm{d}\mathcal{P}^{\mathrm{ref}}}{\mathrm{d}\mathcal{Q}^\theta}(y_{[0,T]}), \tag{11}$$

where the second term can be computed directly from (9) and $y_{[0,T]}$ is obtained by solving (8).

## 2.4 CONTINUOUS-TIME NORMALIZING FLOW

To approximate the log likelihood of DDPM, it was proposed by Song et al. (2021c) to rely on the probability flow ODE. This is an ODE admitting the same marginal distributions $(p_t)_{t \in [0,T]}$ as the noising diffusion given by $\mathrm{d}x_t = -\beta_t\{x_t + \sigma^2 \nabla \ln p_t(x_t)\}\mathrm{d}t$. We use here this ODE to design a continuous-time normalizing flow to sample from $\pi$. For $\theta$ such that $f_\theta(t, x) \approx \nabla \ln \phi_t(x)$ (obtained by minimization of the KL in (10)), we have $x_t + \sigma^2 \nabla \ln p_t(x) \approx \sigma^2 f_\theta(t, x)$. So it is possible to sample approximately from $\pi$ by integrating the following ODE over $[0, T]$

$$\mathrm{d}y_t = \sigma^2 \beta_{T-t} f_\theta(T-t, y_t)\mathrm{d}t, \qquad y_0 \sim \mathcal{N}(0, \sigma^2 I). \tag{12}$$

If denote by $\bar{q}_{T-t}^\theta$ the distribution of $y_t$ then $y_T \sim \bar{q}_0^\theta$ is an approximate sample from $\pi$. We can use this normalizing flow to obtain an unbiased estimate of $Z$ using importance sampling $\hat{Z} = \gamma(y_T)/\bar{q}_0^\theta(y_T)$ for $y_T \sim \bar{q}_0^\theta$. Indeed, contrary to the proposal $q_0^\theta$ induced by (8), we can compute pointwise $\bar{q}_0^\theta$ using the instantaneous change of variables formula (Chen et al., 2018) such that $\ln \bar{q}_0^\theta(y_T) = \ln \mathcal{N}(y_0; 0, \sigma^2 I) - \sigma^2 \int_0^T \beta_{T-t} \nabla \cdot f_\theta(T-t, y_t)\mathrm{d}t$ and where $(y_t)_{t \in [0,T]}$ arises from the ODE (12). The expensive divergence term can be estimated using the Hutchinson estimator (Grathwohl et al., 2018; Song et al., 2021c).

## 2.5 EXTENSION TO UNDERDAMPED LANGEVIN DYNAMICS

For DDPM, it has been proposed to extend the original state $x \in \mathbb{R}^d$ by a momentum variable $p \in \mathbb{R}^d$. One then diffuses the data distribution augmented by a Gaussian distribution on the momentum using an underdamped Langevin dynamics targeting $\mathcal{N}(x; 0, \sigma^2 I)\mathcal{N}(m; 0, M)$ where $M$ is a positive definite mass matrix (Dockhorn et al., 2022). It was demonstrated empirically that the resulting scores are smoother, hence easier to estimate and this leads to improved performance. We adapt here this approach to Monte Carlo sampling; see Section B for more details.

We diffuse $\bar{\pi}(x, m) = \pi(x)\mathcal{N}(m; 0, M)$ using an underdamped Langevin dynamics, i.e.

$$\mathrm{d}x_t = M^{-1}m_t\mathrm{d}t, \quad \mathrm{d}m_t = -\frac{x_t}{\sigma^2}\mathrm{d}t - \beta_t m_t\mathrm{d}t + \sqrt{2\beta_t}M^{1/2}\mathrm{d}B_t, \quad (x_0, m_0) \sim \bar{\pi}. \tag{13}$$

The resulting path measure on $[0, T]$ is denoted $\mathcal{P}$ and the marginal of $(x_t, m_t)$ is denoted $\eta_t$.

Here, the reference process $\mathcal{P}^{\mathrm{ref}}$ is defined by the diffusion (13) initialized using $x_0 \sim \mathcal{N}(0, \sigma^2 I), m_0 \sim \mathcal{N}(0, M)$. This ensures that $\eta_t^{\mathrm{ref}}(x_t, m_t) = \mathcal{N}(x_t; 0, \sigma^2 I)\mathcal{N}(m_t; 0, M)$ for all $t$ and the time-reversal process $(y_t, n_t)_{t \in [0,T]} = (x_{T-t}, m_{T-t})_{t \in [0,T]}$ (where equality is in distribution) of this stationary diffusion satisfies

$$\mathrm{d}y_t = -M^{-1}n_t\mathrm{d}t, \quad \mathrm{d}n_t = \frac{y_t}{\sigma^2}\mathrm{d}t - \beta_{T-t}n_t\mathrm{d}t + \sqrt{2\beta_{T-t}}M^{1/2}\mathrm{d}W_t. \tag{14}$$

Using manipulations identical to Section 2.2, the time-reversal of $\mathcal{P}$ can be also be written for $\phi_t(x, m) := \eta_t(x, m)/\eta_t^{\mathrm{ref}}(x, m)$ as $\mathrm{d}y_t = -M^{-1}n_t\mathrm{d}t$ and

$$\mathrm{d}n_t = \frac{y_t}{\sigma^2}\mathrm{d}t - \beta_{T-t}n_t\mathrm{d}t + 2\beta_{T-t}\nabla_{n_t}\ln\phi_{T-t}(y_t, n_t)\mathrm{d}t + \sqrt{2\beta_{T-t}}M^{1/2}\mathrm{d}W_t. \tag{15}$$

To approximate $\mathcal{P}$, we will consider a parameterized path measure $\mathcal{Q}^\theta$ whose time reversal is defined for $(y_0, n_0) \sim \mathcal{N}(y_0; 0, \sigma^2 I)\mathcal{N}(n_0; 0, M)$ by $\mathrm{d}y_t = -M^{-1}n_t\mathrm{d}t$ and

$$\mathrm{d}n_t = \frac{y_t}{\sigma^2}\mathrm{d}t - \beta_{T-t}n_t\mathrm{d}t + 2\beta_{T-t}Mf_\theta(T-t, y_t, n_t)\mathrm{d}t + \sqrt{2\beta_{T-t}}M^{1/2}\mathrm{d}W_t. \tag{16}$$

We can then express the KL of interest in a compact way similar to Proposition 1, see Appendix B.2. A normalising flow formulation is presented in Appendix B.3.

---

**Algorithm 1** DDS Training

---
1: **Input:** $\sigma > 0, \gamma : \mathbb{R}^d \to \mathbb{R}^+, (\beta_k)_{k=1}^K \in (\mathbb{R}^+)^K, \theta \in \mathbb{R}^p, \lambda > 0$
2: **for** $i = 1, \ldots,$ training iterations **do**
3:     **for** $n = 1, \ldots, N$ **do**     # Iterate over training batch size.
4:         Sample $y_{0,n} \sim \mathcal{N}(0, \sigma^2 I)$ and set $r_0 = 0$
5:         **for** $k = 0, \ldots, K - 1$ **do**     # Iterate over integration steps.
6:             $\lambda_{K-k} := 1 - \sqrt{1 - \alpha_{K-k}}$
7:             $y_{k+1,n} = \sqrt{1 - \alpha_{K-k}} y_{k,n} + 2\sigma^2 \lambda_{K-k} f_\theta(K - k, y_{k,n}) + \sigma\sqrt{\alpha_{K-k}}\varepsilon_{k,n}, \ \varepsilon_{k,n} \overset{\text{i.i.d.}}{\sim} \mathcal{N}(0, I)$
8:             $r_{k+1,n} = r_{k,n} + \frac{2\sigma^2 \lambda_{K-k}^2}{\alpha_{K-k}} \|f_\theta(K - k, y_{k,n})\|^2$
9:         **end for**
10:         $\theta \leftarrow \theta - \lambda \nabla_\theta \frac{1}{N} \sum_{n=1}^N \left( r_{K,n} + \ln\left( \frac{\mathcal{N}(y_{K,n}; 0, \sigma^2 I)}{\pi(y_{K,n})} \right) \right)$
11:     **end for**
12: **end for**
13: **Return:** $\theta$

---

## 3 Denoising Diffusion Samplers: Discrete-Time

We introduce here discrete-time integrators for the SDEs and ODEs introduced in Section 2. Contrary to DDPM, we not only need an integrator for $\mathcal{Q}_\theta$ but also for $\mathcal{P}^{\text{ref}}$ to be able to compute an approximation of the Radon–Nikodym derivative $\mathrm{d}\mathcal{Q}_\theta / \mathrm{d}\mathcal{P}^{\text{ref}}$. Additionally this integrator needs to be carefully designed as explained below to preserve an ELBO. For sake of simplicity, we consider a constant discretization step $\delta > 0$ such that $K = T/\delta$ is an integer. In the indices, we write $k$ for $t_k = k\delta$ to simplify notation; e.g. $x_k$ for $x_{k\delta}$.

### 3.1 Integrator for $\mathcal{P}^{\text{ref}}$

Proposition 1 shows that learning the time reversal in continuous-time can be achieved by minimising the objective $\mathrm{KL}(\mathcal{Q}^\theta \| \mathcal{P})$ given in (10). This expression is obtained by using the identity $\mathrm{KL}(\mathcal{Q}^\theta \| \mathcal{P}) = \mathrm{KL}(\mathcal{Q}^\theta \| \mathcal{P}^{\text{ref}}) + \mathbb{E}_{y_T \sim q_0^\theta}[\ln(p_0^{\text{ref}}(y_T)/p_0(y_T))]$. This is also equivalent to maximizing the corresponding ELBO, $\mathbb{E}_{\mathcal{Q}^\theta}[\ln \hat{Z}]$, for the unbiased estimate $\hat{Z}$ of $Z$ given in (11)

An Euler–Maruyama (EM) discretisation of the corresponding SDEs is obviously applicable. However, it is problematic as established below.

**Proposition 2.** *Consider integrators of $\mathcal{P}^{\text{ref}}$ and $\mathcal{Q}^\theta$ leading to approximations $p^{\text{ref}}(x_{0:K})$ and $q^\theta(x_{0:K})$. Then $\mathrm{KL}(q^\theta \| p^{\text{ref}}) + \mathbb{E}_{y_K \sim q_0^\theta}[\ln(p_0^{\text{ref}}(y_K)/\pi(y_K))]$ is guaranteed to be non-negative and $\mathbb{E}_{\mathcal{Q}^\theta}[\ln \hat{Z}]$ is an ELBO for $\hat{Z} = (\gamma(y_K) p^{\text{ref}}(y_{0:K}))/(\mathcal{N}(y_K; 0, \sigma^2) q^\theta(y_{0:K}))$ if one has $p_K^{\text{ref}}(y_K) = p_0^{\text{ref}}(y_K)$. The EM discretisation of $\mathcal{P}^{\text{ref}}$ does* not *satisfy this condition.*

A direct consequence of this result is that the estimator for $\ln Z$ that uses the EM discretisation can be such that $\mathbb{E}_{\mathcal{Q}^\theta}[\ln \hat{Z}] \geq \ln Z$ as observed empirically in Table 5. To resolve this issue, we derive in the next section an integrator for $\mathcal{P}^{\text{ref}}$ which ensures that $p_k^{\text{ref}}(y) = p_0^{\text{ref}}(y)$ for all $k$.

### 3.2 Ornstein–Uhlenbeck

We can integrate exactly (4). This is given by $y_0 \sim \mathcal{N}(0, \sigma^2 I)$ and

$$y_{k+1} = \sqrt{1 - \alpha_{K-k}} y_k + \sigma\sqrt{\alpha_{K-k}}\varepsilon_k, \quad \varepsilon_k \overset{\text{i.i.d.}}{\sim} \mathcal{N}(0, I), \tag{17}$$

for $\alpha_k = 1 - \exp\left(-2 \int_{(k-1)\delta}^{k\delta} \beta_s \mathrm{d}s\right)^2$. This defines the discrete-time reference process. We propose the following exponential type integrator (De Bortoli, 2022) for (8) initialized using $y_0 \sim \mathcal{N}(0, \sigma^2 I)$

$$y_{k+1} = \sqrt{1 - \alpha_{K-k}} y_k + 2\sigma^2(1 - \sqrt{1 - \alpha_{K-k}}) f_\theta(K - k, y_k) + \sigma\sqrt{\alpha_{K-k}}\varepsilon_k, \tag{18}$$

---

[2] Henceforth, it is assumed that $\alpha_k$ can be computed exactly.

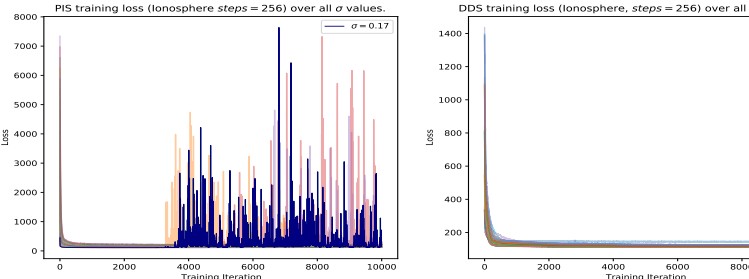

Figure 1: Training loss per hyperparameter: PIS (left) vs DDS (right).

where $\varepsilon_k \stackrel{\text{i.i.d.}}{\sim} \mathcal{N}(0,I)$. Equations (17) and (18) define the time reversals of the reference process $p^{\text{ref}}(x_{0:K})$ and proposal $q^\theta(x_{0:K})$. For its time reversal, we write $q^\theta(y_{0:K}) = \mathcal{N}(y_0; 0, \sigma^2 I) \prod_{k=1}^{K} q^\theta_{k-1|k}(y_{K-k+1}|y_{K-k})$ abusing slightly notation, and similarly for $p^{\text{ref}}(y_{0:K})$. We will be relying on the discrete-time counterpart of (9).

**Proposition 3.** *The log density ratio between $q^\theta(y_{0:K})$ and $p^{\text{ref}}(y_{0:K})$ satisfies for $y_{0:K} \sim q^\theta(y_{0:K})$*

$$\ln\left(\frac{q^\theta(y_{0:K})}{p^{\text{ref}}(y_{0:K})}\right) = 2\sigma^2 \sum_{k=1}^{K} \frac{\lambda_k^2}{\alpha_k} ||f_\theta(k, y_{K-k})||^2 + 2\sigma \sum_{k=1}^{K} \frac{\lambda_k}{\sqrt{\alpha_k}} f_\theta(k, y_{K-k})^\top \varepsilon_k \qquad (19)$$

*where $\lambda_k := 1 - \sqrt{1-\alpha_k}$ and $\varepsilon_k$ defined through (18) is such that $\varepsilon_k \stackrel{\text{i.i.d.}}{\sim} \mathcal{N}(0,I)$. In particular, one obtains from $\text{KL}(q^\theta||p) = \text{KL}(q^\theta||p^{\text{ref}}) + \mathbb{E}_{q_0^\theta}\left[\ln\left(\frac{\pi(y_K)}{\mathcal{N}(y_K;0,\sigma^2 I)}\right)\right]$ that*

$$\text{KL}(q^\theta||p) = \mathbb{E}_{q^\theta}\left[2\sigma^2 \sum_{k=1}^{K} \frac{\lambda_k^2}{\alpha_k} ||f_\theta(k, y_{K-k})||^2 + \ln\left(\frac{\mathcal{N}(y_K;0,\sigma^2 I)}{\pi(y_K)}\right)\right]. \qquad (20)$$

We compute an unbiased gradient of this objective using the reparameterization trick and the JAX software package (Bradbury et al., 2018). The training procedure is summarized in Algorithm 1 in Appendix D. Unfortunately, contrary to DDPM, $q^\theta_{k|K}$ is not available in closed form for $k < K-1$ so we can neither mini-batches over the time index $k$ without having to simulate the process until the minimum sampled time nor reparameterize $x_k$ as in Ho et al. (2020). Once we obtain the parameter $\theta$ minimizing (20), DDS samples from $q^\theta$ using (18). The final sample $y_K$ has a distribution $q_0^\theta$ approximating $\pi$ by design. By using importance sampling, we obtain an unbiased estimate of the normalizing constant $\hat{Z} = (\gamma(y_K)p^{\text{ref}}(y_{0:K}))/(\mathcal{N}(y_K;0,\sigma^2)q^\theta(y_{0:K}))$ for $y_{0:K} \sim q^\theta(y_{0:K})$. Finally Appendices B.4 and B.6 extend this approach to provide a similar approach to discretize the underdamped dynamics proposed in Section 2.5. In this context, the proposed integrators rely on a leapfrog scheme (Leimkuhler & Matthews, 2016).

### 3.3 Theoretical Guarantees

Motivated by DDPM, bounds on the total variation between the target distribution and the distribution of the samples generated by a time-discretization of an approximate time reversal of a forward noising diffusion have been first obtained in (De Bortoli et al., 2021) then refined in (De Bortoli, 2022; Chen et al., 2022; Lee et al., 2022) and extended to the Wasserstein metric. These results are directly applicable to DDS because their proofs rely on assumptions on the score approximation error but not on the way these approximations are learned, i.e. via score matching for DDPM and reverse KL for DDS. For example, the main result in Chen et al. (2022) shows that if the true scores are $L$-Lipschitz, the $L_2(p_t)$ error on the scores is bounded and other mild integrability assumptions then DDS outputs samples $\epsilon$-close in total variation to $\pi$ in $O(L^2 d/\epsilon^2)$ time steps. As pointed out by the authors, this matches state-of-the-art complexity bounds for Langevin Monte Carlo algorithm for sampling targets satisfying a log-Sobolev inequality *without* having to make any log-concavity assumption on $\pi$. However, the assumption on the approximation error for score estimates is less realistic for DDS than DDPM as we do not observe realizations of the forward diffusion.

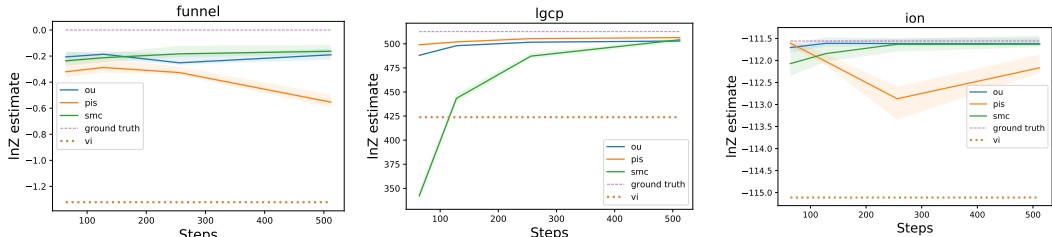

Figure 2: $\ln Z$ estimate (median plus upper/lower quartiles) as a function of number of steps $K$ - a) Funnel , b) LGCP, c) Logistic Ionosphere dataset. Yellow dotted line is MF-VI and dashed magenta is the gold standard.

## 3.4  INTEPRETATION, RELATED WORK AND EXTENSIONS

**DDS as KL and path integral control.** The reverse KL we minimize can be expressed as

$$\text{KL}(q^\theta||p) = \mathbb{E}_{q^\theta}\left[\left\{\ln\left(\frac{\mathcal{N}(x_0;0,\sigma^2 I)}{\pi(x_0)}\right) + \sum_{k=1}^{K}\ln\left(\frac{q^\theta_{k-1|k}(x_{k-1}|x_k)}{p^{\text{ref}}_{k-1|k}(x_{k-1}|x_k)}\right)\right\}\right]. \qquad (21)$$

This objective is a specific example of KL control problem (Kappen et al., 2012). In continuous-time, (10) corresponds to a path integral control problem; see e.g. (Kappen & Ruiz, 2016).

Heng et al. (2020) use KL control ideas so as to sample from a target $\pi$ and estimate $Z$. However, their algorithm relies on $p^{\text{ref}}(x_{0:K})$ being defined by a discretized non-homogeneous Langevin dynamics such that $p_K(x_K)$ is typically not approximating a known distribution. Additionally it approximates the value functions $(\phi_k)_{k=0}^{K}$ using regression using simple linear/quadratic functions. Finally it relies on a good initial estimate of $p(x_{0:K})$ obtained through SMC. This limits the applicability of this methodology to a restricted class of models.

**Connections to Schrödinger Bridges.** The Schrödinger Bridge (SB) problem (Léonard, 2014a; De Bortoli et al., 2021) takes the following form in discrete time. Given a reference density $p^{\text{ref}}(x_{0:K})$, we want to find the density $p^{\text{sb}}(x_{0:K})$ s.t. $p^{\text{sb}} = \arg\min_q\{\text{KL}(q||p^{\text{ref}}) : q_0 = \mu_0,\ q_K = \mu_K\}$ where $\mu_0, \mu_K$ are prescribed distributions. This problem can be solved using iterative proportional fitting (IPF) which is defined by the following recursion with initialization $p^1 = p^{\text{ref}}$

$$p^{2n} = \underset{q}{\arg\min}\{\text{KL}(q||p^{2n-1}) : q_0 = \mu_0\},\ \ p^{2n+1} = \underset{q}{\arg\min}\{\text{KL}(q||p^{2n}) : q_K = \mu_K\}. \qquad (22)$$

Consider the SB problem where $\mu_0(x_0) = \pi(x_0)$, $\mu_K(x_K) = \mathcal{N}(x_K;0,\sigma^2 I)$ and the time-reversal of $p^{\text{ref}}(x_{0:K})$ is defined through (17). In this case, $p^2 = p$ corresponds to the discrete-time version of the noising process and $p^3$ to the time-reversal of $p$ but initialized at $\mu_K$ instead of $p_K$. This is the process DDS is approximating. As $p_K \approx \mu_K$ for $K$ large enough, we have approximately $p^{\text{sb}} \approx p_3 \approx p_2$. We can thus think of DDS as approximating the solution to this SB problem.

Consider now another SB problem where $\mu_0(x_0) = \pi(x_0)$, $\mu_K(x_K) = \delta_0(x_K)$ and $p^{\text{ref}}(x_{0:K}) = \delta_0(x_K)\prod_{k=0}^{K-1}\mathcal{N}(x_k;x_{k+1},\delta\sigma^2 I)$, i.e. $p^{\text{ref}}$ is a pinned Brownian motion running backwards in time. This SB problem was discussed in discrete-time in (Beghi, 1996) and in continuous-time in (Föllmer, 1984; Dai Pra, 1991; Tzen & Raginsky, 2019). In this case, $p^2(x_{0:K}) = \pi(x_0)p^{\text{ref}}(x_{1:K}|x_0)$ is a modified "noising" process that transports $\pi$ to the degenerate measure $\delta_0$ and it is easily shown that $p^{\text{sb}} = p^2$. Sampling from the time-reversal of this measure would generate samples from $\pi$ starting from $\delta_0$. Algorithmically, Zhang et al. (2021) proposed approximating this time-reversal by some projection on some eigenfunctions. In parallel, Barr et al. (2020), Vargas et al. (2021) and Zhang & Chen (2022) approximated this SB by using a neural network parameterization of the gradient of the logarithm of the corresponding value function trained by minimizing a reverse KL. We will adopt the Path Integral Sampler (PIS) terminology proposed by Zhang & Chen (2022) for this approach. DDS can thus be seen as an alternative to PIS which relies on a reference dynamics corresponding to an overdamped or underdamped OU process instead of a pinned Brownian motion. Theoretically the drift of the resulting time-reversal for DDS is not as a steep as for PIS (see Appendix A.2) and empirically this significantly improves numerical stability of the training procedure; see Figure 1. The use of a pinned Brownian motion is also not amenable to the construction of normalizing flows.

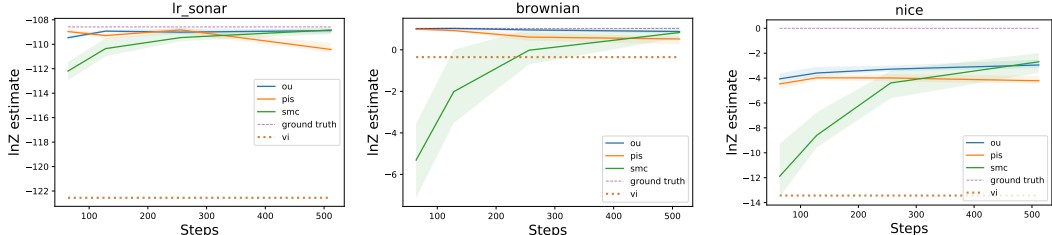

Figure 3: $\ln Z$ estimate as a function of number of steps $K$ - a) Logistic Sonar dataset, b) Brownian motion, c) NICE. Yellow dotted line is MF-VI and dashed magenta is the gold standard.

**Forward KL minimization.** In Jing et al. (2022), diffusion models ideas are also use to sample unnormalized probability densities. The criterion being minimized therein is the forward KL as in DDPM, that is $\mathrm{KL}(p||q^\theta)$. As samples from $p$ are not available, an importance sampling approximation of $p$ based on samples from $q_\theta$ is used to obtain an estimate of this KL and its gradient w.r.t. $\theta$. The method is shown to perform well in low dimensional examples but is expected to degrade significantly in high dimension as the importance sampling approximation will be typically poor in the first training iterations.

## 4 Experiments

We present here experiments for Algorithm 1. In our implementation, $f_\theta$ follows the PIS-GRAD network proposed in (Zhang & Chen, 2022): $f_\theta(k, x) = \mathrm{NN}_1(k, x; \theta) + \mathrm{NN}_2(k; \theta) \odot \nabla \ln \pi(x)$. Across all experiments we use a two layer architecture with 64 hidden units each (for both networks), as in Zhang & Chen (2022), with the exception of the NICE (Dinh et al., 2014) target where we use 3 layers with 512, 256, and 64 units respectively. The final layers are initialised to 0 in order to make the path regularisation term null. We use $\alpha_k^{1/2} \propto \alpha_{\max}^{1/2} \cos^2\left(\frac{\pi}{2} \frac{1-k/K+s}{1+s}\right)$ with $s = 0.008$ as in (Nichol & Dhariwal, 2021). We found that detaching the target score stabilised optimization in both approaches without affecting the final result. We adopt this across experiments, an ablation of this feature can be seen in Appendix C.9.1.

Across all tasks we compare DDS to SMC (Del Moral et al., 2006; Zhou et al., 2016), PIS (Barr et al., 2020; Vargas et al., 2021; Zhang & Chen, 2022), and Mean Field-VI (MF-VI) with a Gaussian variational distribution. We also compare DDS to AIS (Neal, 2001) and optimized variants of AIS using score matching (MCD) (Doucet et al., 2022; Geffner & Domke, 2022) for two standard Bayesian models. Finally we explore a task introduced in (Doucet et al., 2022) that uses a pre-trained normalising flow as a target. Within this setting we propose a benchmarking criterion that allows us to assess mode collapse in high dimensions and explore the benefits of incorporating inductive biases into $f_\theta$. We carefully tuned the hyper-parameters of all algorithms (e.g. step size, diffusion coefficient, and such), details can be found in Appendix C.2. Finally training time can be found in Appendix C.5. Additional experiments for the normalizing flows are presented in Appendix C.11 and for the underdamped approach in Appendix C.12. We note that these extensions did *not* bring any benefit compared to Algorithm 1.

### 4.1 Benchmarking Targets

We first discuss two standard target distributions which are often used to benchmark methods; see e.g. (Neal, 2003; Arbel et al., 2021; Heng et al., 2020; Zhang & Chen, 2022). Results are presented in Figure 2.

**Funnel Distribution:** This 10-dimensional challenging distribution is given by $\gamma(x_{1:10}) = \mathcal{N}(x_1; 0, \sigma_f^2)\mathcal{N}(x_{2:10}; 0, \exp(x_1)I)$, where $\sigma_f^2 = 9$ (Neal, 2003). [3]

---

[3]Zhang & Chen (2022) inadvertently considered the more favourable $\sigma_f = 1$ scenario for PIS but used $\sigma_f = 3$ for other methods. This explains the significant differences between their results and ours.

**Log Gaussian Cox process:** This model arises in spatial statistics (Møller et al., 1998). We use a $d = M \times M = 1600$ grid, resulting in the unnormalized target density $\gamma(x) = \mathcal{N}(x; \mu, K) \prod_{i \in [1:M]^2} \exp(x_i y_i - a \exp(x_i))$.

## 4.2 BAYESIAN MODELS

We explore two standard Bayesian models and compare them with standard AIS and MCD benchmarks (See Appendix C.8) in addition to the SMC and VI benchmarks presented so far. Results for Ionosphere can be found in the 3rd pane of Figure 2 whilst SONAR and Brownian are in Figure 3.

**Logistic Regression:** We set $x \sim \mathcal{N}(0, \sigma_w^2 I), y_i \sim \text{Bernoulli}(\text{sigmoid}(x^\top u_i))$. This Bayesian logistic model is evaluated on two datasets, Ionosphere ($d = 32$) and Sonar ($d = 61$).

**Brownian Motion:** We consider a discretised Brownian motion with a Gaussian observation model and a latent volatility as a time series model, $d = 32$. This model, proposed in the software package developed by Sountsov et al. (2020), is specified in more detail in Appendix C.3.

## 4.3 MODE COLLAPSE IN HIGH DIMENSIONS

**Normalizing Flow Evaluation:** Following Doucet et al. (2022) we train NICE (Dinh et al., 2014) on a down-sampled $d = 14 \times 14$ variant of MNIST (LeCun & Cortes, 2010) and use the trained model as our target. As we can generate samples from our target, we evaluate the methods samples by measuring the Sinkhorn distance between true and estimated samples. This evaluation criteria allows to asses mode collapse for samplers in high dimensional settings. Results can be seen in Table 1 and the third pane of Figure 3.

| Method | DDS | PIS | SMC | MCD |
|---|---|---|---|---|
| $\ln Z$ Estimate | $-3.204 \pm 0.645$ | $-3.933 \pm 0.754$ | $-4.255 \pm 2.043$ | $-6.25$ |
| $\mathcal{W}_{\gamma=0.01}^2(\pi_{\text{true}}, \hat{\pi})$ | $658.079$ | $658.778$ | $750.245$ | NA |

Table 1: Results on NICE target, we performed 30 runs with different seeds for each approach. For MCD we used the results from (Doucet et al., 2022).

## 5 DISCUSSION

We have explored the use of DDPM ideas to sample unnormalized probability distributions and estimate their normalizing constants.

The DDS in Algorithm 1 is empirically competitive with state-of-the-art SMC and numerically more stable than PIS. This comes at the cost of a non-negligible training time compared to SMC. When accounting for it, SMC often provide better performance on simple targets. However, in the challenging multimodal NICE example, even a carefully tuned SMC sampler using Hamiltonian Monte Carlo transitions was not competitive to DDS. This is despite the fact that DDS (and PIS) are prone to mode dropping as any method relying on the reverse KL.

We have also investigated normalizing flows based on the probability flow ODE as well as DDS based on an underdamped dynamics. Our experimental results were disappointing in both cases in high dimensional scenarios. We conjecture that more sophisticated numerical integrators need to be developed for the normalizing flows to be competitive and that the neural network parameterization used in the underdamped scenario should be improved and better leverage the structure of the logarithmic derivative of the value functions.

Overall DDS are a class of algorithms worth investigating and further developing. There has much work devoted to improving successfully DDPM over the past two years including, among many others, modified forward noising mechanisms (Hoogeboom & Salimans, 2022), denoising diffusion implicit models (Song et al., 2021a) and sophisticated numerical integrators (Karras et al., 2022). It is highly likely that some of these techniques will lead to more powerful DDS. Advances in KL and path integral control might also be adapted to DDS to provide better training procedures; see e.g. Thalmeier et al. (2020).

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

## A    APPENDIX

### A.1    PROOF OF PROPOSITION 1

The Radon–Nikodym derivative expression (9) directly follows from an application of Girsanov theorem (see e.g. Klebaner (2012)). The path measures $\mathcal{P}$ and $\mathcal{P}^{\text{ref}}$ are induced by two diffusions following the same dynamics but with different initial conditions so that

$$\begin{aligned}
\mathcal{P}(\omega) &= \mathcal{P}(\omega|y_T)\pi(y_T) \\
&= \mathcal{P}^{\text{ref}}(\omega|y_T)\pi(y_T) \\
&= \mathcal{P}^{\text{ref}}(\omega|y_T)\mathcal{N}(y_T; 0, \sigma^2 I)\frac{\pi(y_T)}{\mathcal{N}(y_T; 0, \sigma^2 I)} \\
&= \mathcal{P}^{\text{ref}}(\omega)\frac{\pi(y_T)}{\mathcal{N}(y_T; 0, \sigma^2 I)}.
\end{aligned}$$

So it follows directly that $\text{KL}(\mathcal{Q}^\theta||\mathcal{P}) = \text{KL}(\mathcal{Q}^\theta||\mathcal{P}^{\text{ref}}) + \mathbb{E}_{y_T \sim q_0^\theta}\left[\ln\left(\frac{p_0^{\text{ref}}(y_T)}{p_0(y_T)}\right)\right]$. Note we apply Girsanov theorem to the time reversals of the path measures $\mathcal{Q}^\theta$ and $\mathcal{P}$ using that the Radon–Nikodym between two path measures and their respective reversals are the same. As $W_t$ is a Brownian motion under $\mathcal{Q}^\theta$, the final expression (10) for $\text{KL}(\mathcal{Q}^\theta||\mathcal{P})$ follows.

### A.2    COMPARING DDS AND PIS DRIFTS FOR GAUSSIAN TARGETS

The optimal drift for both PIS and DDS can be expressed in terms of logarithmic derivative of the value function plus additional terms:

$$b_{\text{DDS}}(x, t) = -\beta_{T-t}(x - 2\sigma^2 \nabla_x \ln\phi_{T-t}(x)), \quad b_{\text{PIS}}(x, t) = \sigma^2 \nabla_x \ln\phi_{T-t}(x) \tag{23}$$

where $\phi_t$ is the corresponding value function for DDS or PIS respectively.

Recall that $\ln\phi_t(x) = \ln p_t(x) - \ln p_t^{\text{ref}}(x)$. For a target $\pi(x) = \mathcal{N}(x; \mu, \Sigma)$, $p_t^{\text{ref}}(x) = \mathcal{N}(x_t; \mu_t^{\text{ref}}, b_t^{\text{ref}} I)$ and $p_{t|0}(x_t|x_0) = \mathcal{N}(x_t; a_t x_0, b_t I)$, we obtain

$$p_t(x) = \int \pi(x_0) p_{t|0}(x|x_0)\mathrm{d}x_0 = \mathcal{N}(x; a_t\mu, a_t^2\Sigma + b_t I)$$

so

$$\nabla \ln\phi_t(x) = -(a_t^2\Sigma + b_t I)^{-1}(x - a_t\mu) + (b_t^{\text{ref}})^{-1}(x - \mu_t^{\text{ref}}). \tag{24}$$

**Corollary 1.** *For DDS, we have* $\mu_t^{\text{ref}} = 0$, $b_t^{\text{ref}} = \sigma^2 I$, $a_t = \sqrt{1 - \lambda_t}$, $b_t = \sigma^2\lambda_t I$ *with* $\lambda_t = 1 - \exp(-2\int_0^t \beta_s \mathrm{d}s)$. *So, for example, for* $\sigma = \beta_t = 1$ *and* $\Sigma = I$, *we have*

$$\nabla \ln\phi_t(x) = \mu\exp(-t), \quad b_{\text{DDS}}(x, t) = -x + 2\mu\exp(-(T - t)). \tag{25}$$

**Corollary 2.** *For PIS, we have a reference process which is is a pinned Brownian motion running backwards in time with* $\mu_t^{\text{ref}} = 0$, $b_t^{\text{ref}} = \sigma^2(T - t)I$, $a_t = \frac{T-t}{T}$, $b_t = \sigma^2\frac{t(T-t)}{T}$. *For* $\sigma = 1, \Sigma = I$, *we obtain*

$$b_{\text{PIS}}(x, t) = -(a_{T-t}^2 + b_{T-t})^{-1}(x - a_{T-t}\mu) + \frac{x}{t}. \tag{26}$$

*In particular* $b_{\text{PIS}}(x, t) \approx \frac{x}{t} - (x - \mu)$ *as* $t \to 0$.

Hence for PIS the drift function explodes close to the origin compared to DDS. This explosion holds for any target $\pi$ as it is only related to the fact that $p_t^{\text{ref}}$ concentrates to $\delta_0$ as $t \to T$. This makes it harder to approximate for a neural network. Additionally when discretizing the resulting diffusion, this means that smaller discretization steps should be used close to $t = 0$.

### A.3 PROOF OF PROPOSITION 2

*Proof.* Let us express the discrete time version of the reference process as

$$p^{\text{ref}}(y_{0:K}) = p_0^{\text{ref}}(y_0) \prod_{k=0}^{K-1} p_{k+1|k}^{\text{ref}}(y_{k+1}|y_k) \tag{27}$$

and denote by $p_k^{\text{ref}}$ the corresponding marginal density of $y_k$ which satisfies

$$p_{k+1}(y_{k+1}) = \int p_{k+1|k}^{\text{ref}}(y_{k+1}|y_k) \ p_k(y_k) \mathrm{d}y_k. \tag{28}$$

The backward decomposition of this joint distribution is given by

$$p^{\text{ref}}(y_{0:K}) = p_K^{\text{ref}}(y_K) \prod_{k=0}^{K-1} p_{k|k+1}^{\text{ref}}(y_k|y_{k+1}), \tag{29}$$

where

$$p_{k|k+1}^{\text{ref}}(y_k|y_{k+1}) = \frac{p_{k+1|k}^{\text{ref}}(y_{k+1}|y_k)p_k(y_k)}{p_{k+1}(y_{k+1})}. \tag{30}$$

If our chosen integrator induces a transition kernel $p_{k+1|k}^{\text{ref}}(y_{k+1}|y_k)$ which is such that $p_K^{\text{ref}}(y_K) = p_0^{\text{ref}}(y_K)$, then

$$p^{\text{ref}}(y_{0:K}) \frac{\pi(y_K)}{p_0^{\text{ref}}(y_K)} = \pi(y_K) \prod_{k=0}^{K-1} p_{k|k+1}^{\text{ref}}(y_k|y_{k+1}) \tag{31}$$

is a valid (normalised) probability density. Hence it follows that

$$\text{KL}(q^\theta(y_{0:K})||p^{\text{ref}}(y_{0:K})) + \mathbb{E}_{y_K \sim q_0^\theta} \left[ \ln \left( \frac{p_0^{\text{ref}}(y_K)}{\pi(y_K)} \right) \right] \tag{32}$$

$$= \text{KL} \left( q^\theta(y_{0:K}) \middle| \middle| p^{\text{ref}}(y_{0:K}) \frac{\pi(y_K)}{p_0^{\text{ref}}(y_K)} \right) \geq 0.$$

If the integrator does not preserve the marginals we have that

$$p^{\text{ref}}(y_{0:K}) \frac{\pi(y_K)}{p_0^{\text{ref}}(y_K)} = \frac{p_K^{\text{ref}}(y_K)}{p_0^{\text{ref}}(y_K)} \pi(y_K) \prod_{k=0}^{K-1} p_{k|k+1}^{\text{ref}}(y_k|y_{k+1}). \tag{33}$$

This is not a probability density and thus the objective is no longer guaranteed to be positive and consequently the expectation of our estimator of $\ln Z$ will not be necessarily a lower bound for $\ln Z$. Finally simple calculations show that the Euler discretisation does not preserve the invariant distribution of $\mathcal{P}^{\text{ref}}$ in DDS. $\qquad \square$

### A.4 PROOF OF PROPOSITION 3

We have

$$\ln \left( \frac{q^\theta(y_{0:K})}{p(y_{0:K})} \right) = \ln \left( \frac{q^\theta(y_{0:K})}{p^{\text{ref}}(y_{0:K})} \right) + \ln \left( \frac{\mathcal{N}(y_0; 0, \sigma^2 I)}{\pi(y_0)} \right). \tag{34}$$

Now by construction, we have from (17) that $p_{k-1|k}^{\text{ref}}(y_{K-k+1}|y_{K-k}) = \mathcal{N}(y_{K-k+1}; \sqrt{1-\alpha_k}y_{K-k}, \sigma^2\alpha_k I)$ and from (18) we obtain $q_{k-1|k}^\theta(y_{K-k+1}|y_{K-k}) = \mathcal{N}(y_{K-k+1}; \sqrt{1-\alpha_k}y_{K-k} + 2\sigma^2(1-\sqrt{1-\alpha_k})f_\theta(k, y_{K-k}), \sigma^2\alpha_k I)$.

It follows that

$$\ln\left(\frac{q^\theta(y_{0:K})}{p^{\text{ref}}(y_{0:K})}\right) \tag{35}$$

$$=\ln\left(\frac{q_K(y_0)}{p_K^{\text{ref}}(y_0)}\right) + \sum_{k=1}^{K}\ln\left(\frac{q_{k-1|k}^\theta(y_{K-k+1}|y_{K-k})}{p_{k-1|k}^{\text{ref}}(y_{K-k+1}|y_{K-k})}\right)$$

$$=\sum_{k=1}^{K}\frac{1}{2\alpha_k\sigma^2}\Big[||y_{K-k+1}-\sqrt{1-\alpha_k}y_{K-k}||^2$$

$$-||y_{K-k+1}-\sqrt{1-\alpha_k}y_{K-k}-2\sigma^2(1-\sqrt{1-\alpha_k})f(k,y_{K-k})||^2\Big],$$

where we have exploited the fact that $q_K(y_0) = p_K^{\text{ref}}(y_0) = \mathcal{N}(y_0; 0, \sigma^2 I)$.

Now using $\epsilon_k := \frac{1}{\sigma\sqrt{\alpha_k}}(y_{K-k+1}-\sqrt{1-\alpha_k}y_{K-k}-2\sigma^2(1-\sqrt{1-\alpha_k})f(k,y_{K-k}))$, we can rewrite (35) as

$$\ln\left(\frac{q^\theta(y_{0:K})}{p^{\text{ref}}(y_{0:K})}\right) \tag{36}$$

$$=\frac{1}{2}\sum_{k=1}^{K}\Big[||\epsilon_k + 2\sigma\frac{(1-\sqrt{1-\alpha_k})}{\sqrt{\alpha_k}}f_\theta(k,y_{K-k})||^2 - ||\epsilon_k||^2\Big]$$

$$=2\sigma^2\sum_{k=1}^{K}\frac{(1-\sqrt{1-\alpha_k})^2}{\alpha_k}||f_\theta(k,y_{K-k})||^2 + 2\sigma\sum_{k=1}^{K}\frac{(1-\sqrt{1-\alpha_k})}{\sqrt{\alpha_k}}f_\theta(k,y_{K-k})^\top\epsilon_k.$$

Now (19) follows directly from (34) and (66). Note we have for $\delta \ll 1$ that $2\sigma^2\frac{(1-\sqrt{1-\alpha_k})^2}{\alpha_k} \approx \sigma^2\beta_k\delta$ and $2\sigma\frac{(1-\sqrt{1-\alpha_k})}{\sqrt{\alpha_k}} \approx \sqrt{2\beta_k\delta\delta}$ as expected from (9).

Finally the final expression (20) of the KL follows now from the fact that $\mathbb{E}_{q^\theta}[f_\theta(k,y_{K-k})^\top\epsilon_k] = \mathbb{E}_{q_k^\theta}[\mathbb{E}_{q_{k-1|k}^\theta}[f_\theta(k,y_{K-k})^\top\epsilon_k|y_{K-k}]] = \mathbb{E}_{q_k^\theta}[f_\theta(k,y_{K-k})^\top\mathbb{E}_{q_{k-1|k}^\theta}[\epsilon_k|y_{K-k}]] = 0$.

## A.5 ALTERNATIVE KULLBACK–LEIBLER DECOMPOSITION

A KL decomposition similar in spirit to the one developed for DDPM (Ho et al., 2020) can be be derived. It leverages the fact that

$$p_{k-1|k,0}(x_{k-1}|x_k,x_0) = \mathcal{N}(x_k;\tilde{\mu}_k(x_k,x_0),\sigma^2\tilde{\beta}_k I) \tag{37}$$

for $\tilde{\mu}_k(x_k,x_0) = \frac{\sqrt{\bar{\alpha}_{k-1}}\beta_k}{1-\bar{\alpha}_k}x_0 + \frac{\sqrt{\alpha_k}(1-\bar{\alpha}_{k-1})}{1-\bar{\alpha}_k}x_k$, $\tilde{\beta}_k = \frac{1-\bar{\alpha}_{k-1}}{1-\bar{\alpha}_k}\beta_k$.

**Proposition 4.** *The reverse Kullback–Leibler discrepancy* $\text{KL}(q^\theta||p)$ *satisfies*

$$\text{KL}(q^\theta||p) = \mathbb{E}_{q^\theta}[\text{KL}(q_K(x_K)||p_{K|0}(x_K|x_0)) + \sum_{k=2}^{K}\text{KL}(q_{k-1|k}^\theta(x_{k-1}|x_k)||p_{k-1|0,k}(x_{k-1}|x_0,x_k))$$

$$+ \text{KL}(q_{0|1}^\theta(x_0|x_1)||\pi(x_0))].$$

*So for* $q_{k-1|k}^\theta(x_{k-1}|x_k) = \mathcal{N}(x_{k-1};\sqrt{1-\beta_k}x_k + \sigma^2\beta_k f_\theta(k,x_k),\sigma^2\beta_k I)$, *the terms* $\text{KL}(q_{k-1|k}^\theta(x_{k-1}|x_k)||p_{k-1|0,k}(x_{k-1}|x_0,x_k))$ *are KL between two Gaussian distributions and can be calculated analytically.*

We found this decomposition to be numerically unstable and prone to diverging in our reverse KL setting.

*Proof.* The reverse KL can be decomposed as follows

$$\mathrm{KL}(q^\theta||p) = \mathbb{E}_{q^\theta}\Big[\ln\Big(\frac{q^\theta(x_{0:K})}{p(x_{0:K})}\Big)\Big] = \mathbb{E}_{q^\theta}\Big[\ln\Big(\frac{q^\theta(x_{0:K})}{\pi(x_0)p(x_{1:K}|x_0)}\Big)\Big]$$

$$= \mathbb{E}_{q^\theta}\Big[\ln\Big(\frac{q^\theta(x_{0:K})}{p(x_{1:K}|x_0)}\Big)\Big] - \mathbb{E}_{q^\theta}[\ln\pi(x_0)]$$

$$= L(\theta) - \mathbb{E}_{q^\theta}[\ln\pi(x_0)]$$

where

$$L(\theta) = \mathbb{E}_{q^\theta}\Big[\ln\Big(\frac{q^\theta(x_{0:K})}{p(x_{1:K}|x_0)}\Big)\Big] = \mathbb{E}_{q^\theta}\Big[\ln q_K(x_K) + \sum_{k=1}^{K}\ln\Big(\frac{q^\theta_{k-1|k}(x_{k-1}|x_k)}{p_{k|k-1}(x_k|x_{k-1})}\Big)\Big]$$

Now using the identity for $k \geq 2$

$$p_{k-1,k|0}(x_{k-1}, x_k|x_0) = p_{k-1|0}(x_{k-1}|x_0)p_{k|k-1}(x_k|x_{k-1})$$
$$= p_{k-1|0,k}(x_{k-1}|x_0, x_k)p_{k|0}(x_k|x_0),$$

we can rewrite $L(\theta)$ as

$$L(\theta) = \mathbb{E}_{q^\theta}\Big[\ln q_K(x_K) + \sum_{k=2}^{K}\ln\Big(\frac{q^\theta_{k-1|k}(x_{k-1}|x_k)}{p_{k-1|0,k}(x_{k-1}|x_0, x_k)}\cdot\frac{p_{k-1|0}(x_{k-1}|x_0)}{p_{k|0}(x_k|x_0)}\Big) + \ln\Big(\frac{q^\theta_{0|1}(x_0|x_1)}{p_{1|0}(x_1|x_0)}\Big)\Big]$$

$$= \mathbb{E}_{q^\theta}\Big[\ln\Big(\frac{q_K(x_K)}{p_{K|0}(x_K|x_0)}\Big) + \sum_{k=2}^{K}\ln\Big(\frac{q^\theta_{k-1|k}(x_{k-1}|x_k)}{p_{k-1|0,k}(x_{k-1}|x_0, x_k)}\Big) + \ln q^\theta_{0|1}(x_0|x_1)\Big]$$

$$= \mathbb{E}_{q^\theta}\Big[\mathrm{KL}(q_K(x_K)||p_{K|0}(x_K|x_0)) + \sum_{k=2}^{K}\mathrm{KL}(q^\theta_{k-1|k}(x_{k-1}|x_k)||p_{k-1|0,k}(x_{k-1}|x_0, x_k))$$

$$+ \ln q^\theta_{0|1}(x_0|x_1)\Big]$$

The result now follows directly. $\qquad\square$

# B  UNDERDAMPED LANGEVIN DYNAMICS

In the generative modeling context, it has been proposed to extend the original state $x \in \mathbb{R}^d$ by a momentum variable $m \in \mathbb{R}^d$. One then diffuses in this extended space using an underdamped Langevin dynamics (Dockhorn et al., 2022) targeting $\mathcal{N}(x; 0, \sigma^2 I)\mathcal{N}(m; 0, M)$. It was demonstrated empirically that the resulting scores one needs to estimate are smoother and this led to improved performance. We adapt this approach to Monte Carlo sampling. This adaptation is non trivial and in particular requires to design carefully numerical integrators.

## B.1  CONTINUOUS TIME

We now consider an augmented target distribution $\pi(x)\mathcal{N}(m; 0, M)$ where $M$ is a positive definite mass matrix. We then diffuse this extended target using the following underdamped Langevin dynamics, i.e.

$$\mathrm{d}x_t = M^{-1}m_t\mathrm{d}t, \tag{38}$$
$$\mathrm{d}m_t = -\frac{x_t}{\sigma^2}\mathrm{d}t - \beta_t m_t\mathrm{d}t + \sqrt{2\beta_t}M^{1/2}\mathrm{d}B_t,$$

where $x_0 \sim \pi, m_0 \sim \mathcal{N}(0, M)$. The resulting path measure on $[0, T]$ is denoted again $\mathcal{P}$. From (Haussmann & Pardoux, 1986), the time-reversal process is also a diffusion satisfying

$$\mathrm{d}y_t = -M^{-1}n_t\mathrm{d}t, \tag{39}$$
$$\mathrm{d}n_t = \frac{y_t}{\sigma^2}\mathrm{d}t + \beta_{T-t}n_t\mathrm{d}t + 2\beta_{T-t}M\nabla_{n_t}\ln\eta_{T-t}(y_t, n_t)\mathrm{d}t + \sqrt{2\beta_{T-t}}M^{1/2}\mathrm{d}W_t,$$

for $(y_0, n_0) \sim \eta_T$ where $\eta_t$ denotes the density of $(x_t, m_t)$ under (38).

Now consider a reference process $\mathcal{P}^{\text{ref}}$ on $[0, T]$ defined by the forward process (38) initialized using $x_0 \sim \mathcal{N}(0, \sigma^2), m_0 \sim \mathcal{N}(0, M)$. In this case one can check that $\eta_t^{\text{ref}}(x_t, m_t) = \mathcal{N}(x_t; 0, \sigma^2 I)\mathcal{N}(m_t; 0, M)$ and the time-reversal process of $\mathcal{P}^{\text{ref}}$ satisfies

$$\mathrm{d}y_t = -M^{-1}n_t\mathrm{d}t, \tag{40}$$

$$\mathrm{d}n_t = \frac{y_t}{\sigma^2}\mathrm{d}t + \beta_{T-t}n_t\mathrm{d}t - 2\beta_{T-t}n_t\mathrm{d}t + \sqrt{2\beta_{T-t}}M^{1/2}\mathrm{d}W_t$$

$$= \frac{y_t}{\sigma^2}\mathrm{d}t - \beta_{T-t}n_t\mathrm{d}t + \sqrt{2\beta_{T-t}}M^{1/2}\mathrm{d}W_t,$$

as $\nabla_n \ln \eta_t^{\text{ref}}(y, n) = \nabla_n \ln(\mathcal{N}(y; 0, \sigma^2)\mathcal{N}(n; 0, M)) = -M^{-1}n$.

Hence it follows that the time-reversal (39) of $\mathcal{P}$ can be also be written as

$$\mathrm{d}y_t = -M^{-1}n_t\mathrm{d}t, \tag{41}$$

$$\mathrm{d}n_t = \frac{y_t}{\sigma^2}\mathrm{d}t - \beta_{T-t}n_t\mathrm{d}t + 2\beta_{T-t}M\nabla_{n_t} \ln \phi_{T-t}(y_t, n_t)\mathrm{d}t + \sqrt{2\beta_{T-t}}M^{1/2}\mathrm{d}W_t,$$

where $\phi_t(x, m) := \eta_t(x, m)/\eta_t^{\text{ref}}(x, m)$.

To approximate $\mathcal{P}$, we consider a parameterized path measure $\mathcal{Q}^\theta$ whose time reversal is defined for $(y_0, n_0) \sim \mathcal{N}(y_0; 0, I)\mathcal{N}(n_0; 0, M)$ by $\mathrm{d}y_t = -M^{-1}n_t\mathrm{d}t$ and

$$\mathrm{d}n_t = \frac{y_t}{\sigma^2}\mathrm{d}t - \beta_{T-t}n_t\mathrm{d}t + 2\beta_{T-t}Mf_\theta(T - t, y_t, n_t)\mathrm{d}t + \sqrt{2\beta_{T-t}}M^{1/2}\mathrm{d}W_t. \tag{42}$$

### B.2 LEARNING THE TIME-REVERSAL THROUGH KL MINIMIZATION

To approximate $\mathcal{P}$, we will consider a parameterized diffusion whose time reversal is defined by

$$\mathrm{d}y_t = -M^{-1}n_t\mathrm{d}t, \tag{43}$$

$$\mathrm{d}n_t = \frac{y_t}{\sigma^2}\mathrm{d}t - \beta_{T-t}n_t\mathrm{d}t + 2\beta_{T-t}Mf_\theta(T - t, y_t, n_t)\mathrm{d}t + \sqrt{2\beta_{T-t}}M^{1/2}\mathrm{d}W_t$$

for $(y_0, n_0) \sim \mathcal{N}(y_0; 0, I)\mathcal{N}(n_0; 0, M)$ inducing a path measure $\mathcal{Q}^\theta$ on the time interval $[0, T]$.

We can now compute the Radon-Nikodym derivative between $\mathcal{Q}^\theta$ and $\mathcal{P}^{\text{ref}}$ using an extension of Girsanov theorem (Theorem A.3. in Sottinen & Särkkä (2008))

$$\ln \frac{\mathrm{d}\mathcal{Q}^\theta}{\mathrm{d}\mathcal{P}^{\text{ref}}} = \int_0^T \sqrt{2\beta_{T-t}}M^{1/2}f_\theta(T - t, y_t, n_t)^\top\mathrm{d}W_t + \frac{1}{2}\int_0^T ||2\beta_{T-t}Mf_\theta(T - t, y_t, n_t)||^2_{(2\beta_{T-t}M)^{-1}}\mathrm{d}t \tag{44}$$

To summarize, we have the following proposition.

**Proposition 5.** *The Radon-Nikodym derivative $\frac{\mathrm{d}\mathcal{Q}^\theta}{\mathrm{d}\mathcal{P}^{\text{ref}}}(y_{[0,T]}, n_{[0,T]})$ satisfies under $\mathcal{Q}^\theta$*

$$\ln\left(\frac{\mathrm{d}\mathcal{Q}^\theta}{\mathrm{d}\mathcal{P}^{\text{ref}}}\right) = \int_0^T \beta_{T-t}||f_\theta(T - t, y_t, n_t)||^2_M\mathrm{d}t + \int_0^T \sqrt{2\beta_{T-t}}M^{1/2}f_\theta(T - t, y_t, n_t)^\top\mathrm{d}W_t. \tag{45}$$

*From $\mathrm{KL}(\mathcal{Q}^\theta||\mathcal{P}) = \mathrm{KL}(\mathcal{Q}^\theta||\mathcal{P}^{\text{ref}}) + \mathbb{E}_{Q^\theta}\left[\ln\left(\frac{p_0^{\text{ref}}(y_T, n_T)}{\eta_0(y_T, n_T)}\right)\right]$, it follows that*

$$\mathrm{KL}(\mathcal{Q}^\theta||\mathcal{P}) = \mathbb{E}_{\mathcal{Q}^\theta}\left[\int_0^T \beta_{T-t}||f_\theta(T - t, y_t, n_t)||^2_M\mathrm{d}t + \ln\left(\frac{\mathcal{N}(y_T; 0, \sigma^2 I)}{\pi(y_T)}\right)\right]. \tag{46}$$

The second term on the r.h.s. of (46) follows from the fact that $\ln(p_0^{\text{ref}}(y_T, n_T)/p_0(y_T, n_T)) = \ln(\mathcal{N}(y_T; 0, \sigma^2 I)/\pi(y_T))$.

### B.3 NORMALIZING FLOW THROUGH ORDINARY DIFFERENTIAL EQUATION

The following ODE gives exactly the same marginals $\eta_T$ as the SDE (38) defining $\mathcal{P}$

$$\mathrm{d}x_t = M^{-1}m_t\mathrm{d}t, \tag{47}$$

$$\mathrm{d}m_t = -\frac{x_t}{\sigma^2}\mathrm{d}t - \beta_t m_t\mathrm{d}t - \beta_t M\nabla_{m_t}\ln\eta_t(x_t, m_t)\mathrm{d}t,$$

$$= -\frac{x_t}{\sigma^2}\mathrm{d}t - \beta_t M\nabla_{m_t}\ln\phi_t(x_t, m_t)\mathrm{d}t,$$

Thus if we integrate its time reversal from 0 to $T$ starting from $(y_0, q_0) \sim \eta_T$

$$\mathrm{d}y_t = -M^{-1}n_t\mathrm{d}t, \tag{48}$$

$$\mathrm{d}n_t = \frac{y_t}{\sigma^2}\mathrm{d}t + \beta_{T-t}M\nabla_{n_t}\ln\phi_{T-t}(y_t, n_t)\mathrm{d}t,$$

then we would obtain at time $T$ a sample $(y_T, n_T) \sim \pi(y_T)\mathcal{N}(n_T; 0, M)$.

In practice, this suggests that once we have learned an approximation $f_{\theta^*}(t, x, m) \approx \nabla_m \ln\phi_t(x, m)$ by minimization of the ELBO then we can construct a proposal using

$$\mathrm{d}y_t = -M^{-1}n_t\mathrm{d}t, \tag{49}$$

$$\mathrm{d}n_t = \frac{y_t}{\sigma^2}\mathrm{d}t + \beta_{T-t}Mf_{\theta^*}(T - t, y_t, n_t)\mathrm{d}t,$$

for $(y_0, n_0) \sim \mathcal{N}(y_0; 0, \sigma^2 I)\mathcal{N}(n_0; 0, M)$. The resulting sample $(y_T, n_T) \sim \bar{\eta}_0$ will have distribution close to $\pi \times \mathcal{N}(0, M)$. Again it is possible to compute pointwise the distribution of this sample to perform an importance sampling correction.

### B.4 FROM CONTINUOUS TIME TO DISCRETE TIME

We now need to come up with discrete-time integrator for the time-reversal of $\mathcal{P}^{\mathrm{ref}}$ given by (40) and the time-reversal of $\mathcal{Q}^\theta$ given by (43). Let us start with (40). We split it into the two components

$$\mathrm{d}y_t = -M^{-1}n_t\mathrm{d}t, \qquad \mathrm{d}n_t = \frac{y_t}{\sigma^2}\mathrm{d}t, \tag{50}$$

and

$$\mathrm{d}y_t = 0, \qquad \mathrm{d}n_t = -\beta_{T-t}n_t\mathrm{d}t + \sqrt{2\beta_{T-t}}M^{1/2}\mathrm{d}W_t. \tag{51}$$

We will compose these transitions. To obtain $(y_{k+1}, n_{k+1})$ from $(y_k, n_k)$, we first integrate (50). To do this, consider the Hamiltonian equation $\mathrm{d}y_t = M^{-1}n_t\mathrm{d}t, \mathrm{d}n_t = -\frac{y_t}{\sigma^2}\mathrm{d}t$ which preserves $\mathcal{N}(y; 0, \sigma^2 I)\mathcal{N}(n; 0, M)$ as invariant distribution. We can integrate this ODE exactly over an interval of length $\delta$ and denote its solution $\Phi(y, n)$; see Section B.5 for details. We use its inverse $\Phi^{-1}(y, n) = \Phi_{\mathrm{flip}}(y, n) \circ \Phi(y, n) \circ \Phi_{\mathrm{flip}}(y, n)$ where $\Phi_{\mathrm{flip}}(y, n) = (y, -n)$ so that $(y_{k+1}, n'_k) = \Phi^{-1}(y_k, n_k)$. Then we integrate exactly (51) using

$$n_{k+1} = \sqrt{1 - \alpha_{K-k}}n'_k + \sqrt{\alpha_{k-1}}M^{1/2}\epsilon_k. \tag{52}$$

We have thus design a transition of the form

$$p^{\mathrm{ref}}(y_{k+1}, n_{k+1}, n'_k|y_k, n_k) = \delta_{\Phi^{-1}(y_k, n_k)}(y_{k+1}, n'_k)\mathcal{N}(n_{k+1}; \sqrt{1 - \alpha_{K-k}}n'_k; \alpha_{K-k}M). \tag{53}$$

Now to integrate (43), we split it in three parts. We first integrate (50) using exactly the same integrator $(y_{k+1}, n'_k) = \Phi^{-1}(y_k, n_k)$. Then we integrate

$$\mathrm{d}n_t = 2\beta_{T-t}Mf_\theta(T - t, y_t, n_t)\mathrm{d}t \tag{54}$$

using

$$n''_k = n'_k + 2(1 - \sqrt{1 - \alpha_{K-k}})Mf_\theta(K - k, y_{k+1}, n'_k), \tag{55}$$

where we abuse notation and write $f_\theta(K - k, y_{k+1}, n'_k)$ instead of $f_\theta((K - k)\delta, y_{k+1}, n'_k)$. Finally, we integrate the OU part using (52) but replacing $n'_k$ by with $n''_k$. So the final transition we get is

$$q^\theta(y_{k+1}, n_{k+1}, n'_k, n''_k|y_k, n_k) = \delta_{\Phi^{-1}(y_k, n_k)}(y_{k+1}, n'_k)\delta_{n'_k + 2(1 - \sqrt{1 - \alpha_{K-k}})Mf_\theta(K-k, y_{k+1}, n'_k)}(n''_k)$$

$$\times \mathcal{N}(n_{k+1}; \sqrt{1 - \alpha_{K-k}}n''_k; \alpha_{K-k}M).$$

However, we can integrate $n''_k$ analytically to obtain

$$q^\theta(y_{k+1}, n_{k+1}, n'_k|y_k, n_k) = \delta_{\Phi^{-1}(y_k, n_k)}(y_{k+1}, n'_k)$$

$$\times \mathcal{N}(n_{k+1}; \sqrt{1 - \alpha_{K-k}}(n'_k + 2(1 - \sqrt{1 - \alpha_{K-k}})Mf_\theta(K - k, y_{k+1}, n'_k)); \alpha_{K-k}M). \tag{56}$$

### B.5 EXACT SOLUTION OF HAMILTONIAN EQUATION

Consider the Hamiltonian equation and $M = I$, the solution of

$$\mathrm{d}x_t = m_t \mathrm{d}t, \qquad \mathrm{d}m_t = -\frac{x_t}{\sigma^2}\mathrm{d}t \tag{57}$$

can be exactly written as $(x_t, m_t) = \Phi_t(x_0, m_0)$ defined through

$$x_t = x_0 \cos(t/\sigma) + m_0 \sigma \sin(t/\sigma), \quad m_t = -\frac{x_0}{\sigma}\sin(t/\sigma) + m_0 \cos(t/\sigma). \tag{58}$$

This is the so-called harmonic oscillator.

Now the inverse of the Hamiltonian flow satisfies $\Phi^{-1}(x, m) = \Phi_{\mathrm{flip}}(x, m) \circ \Phi(x, m) \circ \Phi_{\mathrm{flip}}(x, m)$ (see e.g. Leimkuhler & Matthews (2016)) so we can simply integrate

$$\mathrm{d}y_t = -n_t \mathrm{d}t, \qquad \mathrm{d}n_t = \frac{y_t}{\sigma^2}\mathrm{d}t, \tag{59}$$

using $(y_{k+1}, n'_k) = \Phi_\tau^{-1}(y_k, n_k)$.

This gives

$$y_{k+1} = y_k \cos(\tau/\sigma) - n_k \sigma \sin(\tau/\sigma) \tag{60}$$

$$n'_k = -(-\frac{y_k}{\sigma}\sin(\tau/\sigma) - n_k \cos(t/\sigma))$$

$$= \frac{y_k}{\sigma}\sin(\tau/\sigma) + n_k \cos(t/\sigma). \tag{61}$$

Obviously, if we have $M = \alpha I$ then

$$\mathrm{d}x_t = \frac{m_t}{\alpha}\mathrm{d}t, \qquad \mathrm{d}m_t = -\frac{x_t}{\sigma^2}\mathrm{d}t, \tag{62}$$

then we use a reparameterization $\tilde{m}_t = m_t/\alpha$ and

$$\mathrm{d}x_t = \tilde{m}_t \mathrm{d}t, \qquad \mathrm{d}\tilde{m}_t = -\frac{1}{\alpha}\frac{x_t}{\sigma^2}\mathrm{d}t, \tag{63}$$

and the solution is as above with $\sigma$ being replaced by $\sigma\sqrt{\alpha}$.

So for $\tilde{n}_k = n_k/\alpha$, we have

$$y_{k+1} = y_k \cos(\tau/(\sqrt{\alpha}\sigma)) - \tilde{n}_k \sigma\sqrt{\alpha}\sin(\tau/(\sqrt{\alpha}\sigma)),$$

$$\tilde{n}'_k = \frac{y_k}{\sqrt{\alpha}\sigma}\sin(\tau/(\sqrt{\alpha}\sigma)) + \tilde{n}_k \cos(t/(\sqrt{\alpha}\sigma)),$$

so as $n_k = \alpha\tilde{n}_k$ and similarly $n'_k = \alpha\tilde{n}'_k$, we finally obtain

$$y_{k+1} = y_k \cos(\tau/(\sqrt{\alpha}\sigma)) - n_k \frac{\sigma}{\sqrt{\alpha}}\sin(\tau/(\sqrt{\alpha}\sigma)), \tag{64}$$

$$n'_k = \frac{\sqrt{\alpha}}{\sigma}y_k \sin(\tau/(\sqrt{\alpha}\sigma)) + n_k \cos(\tau/(\sqrt{\alpha}\sigma)). \tag{65}$$

### B.6 DERIVATION OF UNDERDAMPENED KL IN DISCRETE TIME

In this section we derive the discrete-time KL for the underdamped noising dynamics.

**Proposition 6.** *The log density ratio* $\mathrm{lr} = \ln(q^\theta(y_{0:K}, n_{0:K}, n'_{0:K})/p^{\mathrm{ref}}(y_{0:K}, n_{0:K}, n'_{0:K}))$ *for* $y_{0:K}, n_{0:K}, n'_{0:K} \sim q^\theta(\cdot)$ *equals*

$$\mathrm{lr} = \sum_{k=1}^K 2\Big[\frac{\kappa_k^2}{\alpha_k}||f_\theta(k, y_{K-k+1}, n'_{K-k})||_M^2 + M^{1/2}\frac{\kappa_k}{\sqrt{\alpha_k}}f_\theta(k, y_{K-k+1}, n'_{K-k})^\top \varepsilon_k\Big] \tag{66}$$

*where* $\kappa_k := \sqrt{1-\alpha_k}(1-\sqrt{1-\alpha_k})$ *and* $\varepsilon_k$ *is obtained as a function of* $n_{k+1}$ *and* $n''_k$ *when describing the integrator for* $Q^\theta$. *In particular, we have* $\varepsilon_k \overset{\mathrm{i.i.d.}}{\sim} \mathcal{N}(0, I)$ *and one obtains*

$$\mathrm{KL}(q^\theta||p) = 2\mathbb{E}_{q^\theta}\left[\sum_{k=1}^K \frac{\kappa_k^2}{\alpha_k}\Big[||f_\theta(k, y_{K-k+1}, n'_{K-k})||_M^2 + \ln\Big(\frac{\mathcal{N}(y_K; 0, \sigma^2 I)}{\pi(y_K)}\Big)\Big]\right]. \tag{67}$$

The integrator has been designed so that the ratio between the transitions of the proposal (56) and the reference process (53) is well-defined as the deterministic parts are identical in the two transitions so cancel and

$$\frac{q^\theta(y_{k+1}, n_{k+1}, n'_k | y_k, n_k)}{p^{\text{ref}}(y_{k+1}, n_{k+1}, n'_k | y_k, n_k)} \tag{68}$$

$$= \frac{\mathcal{N}(n_{k+1}; \sqrt{1 - \alpha_{K-k}}(n'_k + 2(1 - \sqrt{1 - \alpha_{K-k}})M f_\theta(K - k, y_{k+1}, n'_k)); \alpha_{K-k} M)}{\mathcal{N}(n_{k+1}; \sqrt{1 - \alpha_{K-k}} n'_k; \alpha_{K-k} M)}.$$

The calculations to compute the Radon–Nikodym derivative are now very similar to what we did in the proof of Proposition 3.

We have

$$\ln\left(\frac{q^\theta(y_{0:K}, n_{0:K}, n'_{0:K})}{p^{\text{ref}}(y_{0:K}, n_{0:K}, n'_{0:K})}\right) \tag{69}$$

$$= \ln\left(\frac{q_K(y_0, n_0)}{p^{\text{ref}}_K(y_0, n_0)}\right) + \sum_{k=1}^{K} \ln\left(\frac{q^\theta_{k-1|k}(y_{K-k+1}, n_{K-k+1}, n'_{K-k} | y_{K-k}, n_{K-k})}{p^{\text{ref}}_{k-1|k}(y_{K-k+1}, n_{K-k+1}, n'_{K-k} | y_{K-k}, n_{K-k})}\right)$$

$$= \sum_{k=1}^{K} \Big[ \|n_{K-k+1} - \sqrt{1 - \alpha_k} n'_{K-k}\|^2_{(2\alpha_k M)^{-1}}$$

$$- \|n_{K-k+1} - \sqrt{1 - \alpha_k}(n'_{K-k} + 2(1 - \sqrt{1 - \alpha_k})M f_\theta(k, y_{K-k+1}, n'_{K-k}))\|^2_{(2\alpha_k M)^{-1}} \Big]$$

where we have exploited the fact that $q_K(y_0, n_0) = p^{\text{ref}}_K(y_0, n_0) = \mathcal{N}(y_0; 0, \sigma^2 I)\mathcal{N}(n_0; 0, M)$.

Now let us introduce

$$\epsilon_k := \frac{M^{-1/2}}{\sqrt{\alpha_k}}(n_{K-k+1} - \sqrt{1 - \alpha_k} n'_{K-k} - 2\sqrt{1 - \alpha_k}(1 - \sqrt{1 - \alpha_k})M f_\theta(k, y_{K-k}, n'_{K-k})) \tag{70}$$

hence

$$\frac{M^{-1/2}}{\sqrt{\alpha_k}}(n_{K-k+1} - \sqrt{1 - \alpha_k} n'_{K-k})$$

$$= \epsilon_k + \frac{2M^{-1/2}}{\sqrt{\alpha_k}}\sqrt{1 - \alpha_k}(1 - \sqrt{1 - \alpha_k})M f_\theta(k, y_{K-k+1}, n'_{K-k})$$

$$= \epsilon_k + \frac{2M^{1/2}}{\sqrt{\alpha_k}}\sqrt{1 - \alpha_k}(1 - \sqrt{1 - \alpha_k}) f_\theta(k, y_{K-k+1}, n'_{K-k}).$$

We can rewrite (69) as

$$\ln\left(\frac{q^\theta(y_{0:K}, n_{0:K}, n'_{0:K})}{p^{\text{ref}}(y_{0:K}, n_{0:K}, n'_{0:K})}\right) \tag{71}$$

$$= \frac{1}{2} \sum_{k=1}^{K} \Big[ \|\epsilon_k + \frac{2M^{1/2}}{\sqrt{\alpha_k}}\sqrt{1 - \alpha_k}(1 - \sqrt{1 - \alpha_k}) f_\theta(k, y_{K-k+1}, n'_{K-k}))\|^2 - \|\epsilon_k\|^2 \Big]$$

$$= \frac{1}{2} \sum_{k=1}^{K} \Big[ \|f_\theta(k, y_{K-k+1}, n'_{K-k}))\|^2_{4M \frac{(1-\alpha_k)(1-\sqrt{1-\alpha_k})^2}{\alpha_k}}$$

$$+ \frac{4M^{1/2}}{\sqrt{\alpha_k}}\sqrt{1 - \alpha_k}(1 - \sqrt{1 - \alpha_k}) f_\theta(k, y_{K-k+1}, n'_{K-k}))^\top \varepsilon_k \Big]$$

$$= \sum_{k=1}^{K} \Big[ \|f_\theta(k, y_{K-k+1}, n'_{K-k}))\|^2_{2M \frac{(1-\alpha_k)(1-\sqrt{1-\alpha_k})^2}{\alpha_k}}$$

$$+ 2M^{1/2}\frac{\sqrt{1 - \alpha_k}(1 - \sqrt{1 - \alpha_k})}{\sqrt{\alpha_k}} f_\theta(k, y_{K-k+1}, n'_{K-k}))^\top \varepsilon_k \Big]$$

where we note that $\frac{\sqrt{1-\alpha_k}(1-\sqrt{1-\alpha_k})}{\sqrt{\alpha_k}} \approx \alpha_k/2 \approx \beta_{k\delta}\delta$.

## C    EXPERIMENTS

In this section we incorporate additional experiments and ablations as well as further experimental detail. For both PIS and DDS we created a grid with $\delta = \frac{K}{T} = 0.05$ and values of $T \in \{3.4, 6.4, 12.8, 25.6\}$ and corresponding number of steps $K \in \{64, 128, 128, 256\}$. When tuning PIS we explore different values of $\sigma$ which in practice, when discretised, amounts to the same effect as changing $\delta$. For DDS we apply the cosine schedule directly on the number of steps however we ensure that $\sum_k \alpha_k = \alpha_{\max} T$ such that we have a similar scaling as in PIS. Finally both PIS and DDS where trained with Adam (Kingma & Ba, 2015) to at most 11000 iterations, although in most cases most both converged in less than 6000 iterations. Across all experiments used a sample size of 300 to estimate the ELBO at train time and of 2000 to for the reported IS estimator of $Z$ for all methods.

Finally for the ground truth where available we use the true normalizing constant as is the case with the Funnel distribution and the Normalising Flows used in the NICE Dinh et al. (2014) task. For the rest of the tasks we follow prior work (Zhang & Chen, 2022; Arbel et al., 2021) use a long run SMC chain (1000 temperatures, 30 seeds with 2000 samples each).

### C.1    NETWORK PARAMETRISATION

In order to improve numerical stability we re-scale the neural network drift parametrisation as follows:

$$
f_\theta(k, y) = 2^{-1} \lambda_K^{-1} \alpha_k \tilde{f}_\theta(k, y),
$$
$$
\tilde{f}_\theta(k, y) = \mathrm{NN}_1(k, y; \theta) + \mathrm{NN}_2(k; \theta) \odot \nabla \ln \pi(y).
$$

This was done such that the reciprocal term $\alpha_{K-k}^{-1}$ in the DDS objective is cancelled as the term reaches very small values in the boundaries causing the overall objective to be large. Finally as $\lambda_k = 1 - \sqrt{1 - \alpha_k} \approx \frac{\alpha_k}{2}$ it follows that our proposed re-parametrisation converges to the same SDE as before, but now with stabler and simpler updates:

$$
y_{k+1,n} = \sqrt{1 - \alpha_{K-k}} y_{k,n} + \sigma^2 \alpha_{K-k} \tilde{f}_\theta(K - k, y_{k,n}) + \sigma \sqrt{\alpha_{K-k}} \varepsilon_{k,n}, \quad \varepsilon_{k,n} \overset{\text{i.i.d.}}{\sim} \mathcal{N}(0, I), \quad (72)
$$
$$
r_{k+1,n} = r_{k,n} + 2^{-1} \sigma^2 \alpha_{K-k} || \tilde{f}_\theta(K - k, y_{k,n}) ||^2. \tag{73}
$$

### C.2    TUNING HYPER-PARAMETERS

For both DDS and PIS we explore a grid of 25 hyper-parameter values. For PIS we search for the best performing value of the volatility coefficient over 25 values, depending on the task we vary the end points of the drift however we noticed that PIS leads to numerical instabilities for $\gamma > 4$ thus we never search for values larger than this. For DDS we searched across 5 values for $\sigma$ and 5 values for $\alpha_{\max}$, this led to a total of 25 combinations we explored for each experiment. Finally for SMC we searched over 3 of its different step sizes leading to a total of $5^3 = 125$.

### C.3    BROWNIAN MOTION MODEL WITH GAUSSIAN OBSERVATION NOISE

The model is given by

$$
\begin{aligned}
\alpha_{\mathrm{inn}} &\sim \mathrm{LogNormal}(0, 2), \\
\alpha_{\mathrm{obs}} &\sim \mathrm{LogNormal}(0, 2), \\
x_1 &\sim \mathcal{N}(0, \alpha_{\mathrm{inn}}), \\
x_i &\sim \mathcal{N}(x_{i-1}, \alpha_{\mathrm{inn}}), \quad i = 2, \ldots 20, \\
y_i &\sim \mathcal{N}(x_i, \alpha_{\mathrm{obs}}), \quad i = 1, \ldots 30.
\end{aligned}
$$

The goal is to perform inference over the variables $\alpha_{\mathrm{inn}}, \alpha_{\mathrm{obs}}$ and $\{x_i\}_{i=1}^{30}$ given the observations $\{y_i\}_{i=1}^{10} \cup \{y_i\}_{i=20}^{30}$.

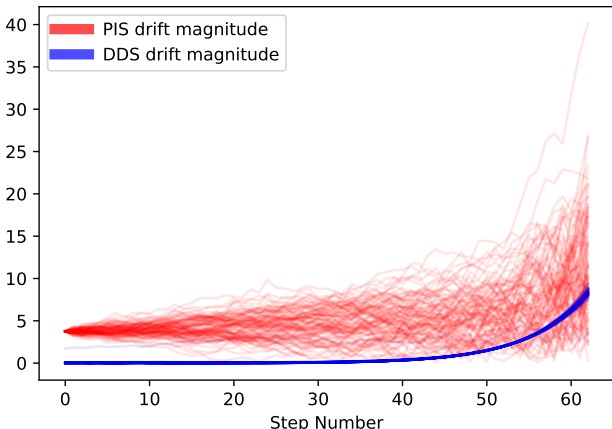

Figure 4: Magnitude of the learnt neural net approximation of the drift $\nabla_x \ln \phi_t(x)$ (see (23)) as a function of $t$.

| | Training Times per Iteration - seconds (LGCP) | | | |
| --- | --- | --- | --- | --- |
| | $k = 64$ | $k = 128$ | $k = 256$ | $k = 512$ |
| DDS | $0.104 \pm 0.0004$ | $0.205 \pm 0.0004$ | $0.406 \pm 0.0004$ | $0.809 \pm 0.0004$ |
| PIS | $0.104 \pm 0.0004$ | $0.205 \pm 0.0003$ | $0.406 \pm 0.0005$ | $0.808 \pm 0.0006$ |

Table 2: Training time per ELBO gradient update on 300 samples.

## C.4 DRIFT MAGNITUDE

In Figure 4 we compare the magnitudes of the NN drifts learned by PIS and DDS on a scalar target $\mathcal{N}(6, \sigma^2)$ target. The drifts where sampled on randomly evaluated trajectories from each of the learned samplers. We observe the PIS drift to have large magnitudes with higher variance more prone to numerical instabilities, whilst the OU drift has a notably lower magnitude with significantly less variance. We conjecture this is the reason that PIS becomes unstable at training time for a large number of steps across several of our PIS experiments, many of which did not converge due to numerical instabilities. For simplicity in this experiment we set $\alpha_k$ to be uniform.

## C.5 TRAINING TIME

We evaluate the training times of DDS and PIS on an 8-chip TPU circuit and average over 99 runs. Table 2 shows running times per training iteration. In total we trained for 11000 iterations thus the total training times are 19 minutes; 37 minutes; 1 hour and 14 minutes; 2 hours and 26 minutes for $k = 64, 128, 256, 512$ respectively.

## C.6 VARIATIONAL ENCODER IN LATENT SPACE

Following Arbel et al. (2021) we explore estimating the posterior of a pre-trained Variational Auto Encoder (VAE) (Kingma & Welling, 2013). We found all approaches to reach a very small error for $\ln Z$ very quickly however as seen with PIS across experiments results became unstable for large $T$ and thus many hyper-parameters when tuning the volatility for PIS led to the loss exploding.

## C.7 GENERATED IMAGES

In this section we provide some of the generated samples for the normalising flow evaluation task. In Figure 6 we can observe how both PIS and DDS are able to generate more image classes than SMC due to mode collapse. Out of the 3 approaches we can see that DDS mode collapses the least whilst SMC generates the higher quality images. Using a neural network with inductive biases such

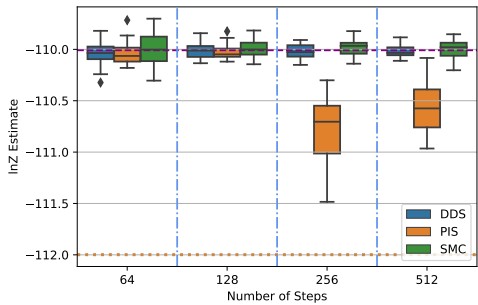

Figure 5: Results on pretrained VAE from Arbel et al. (2021).

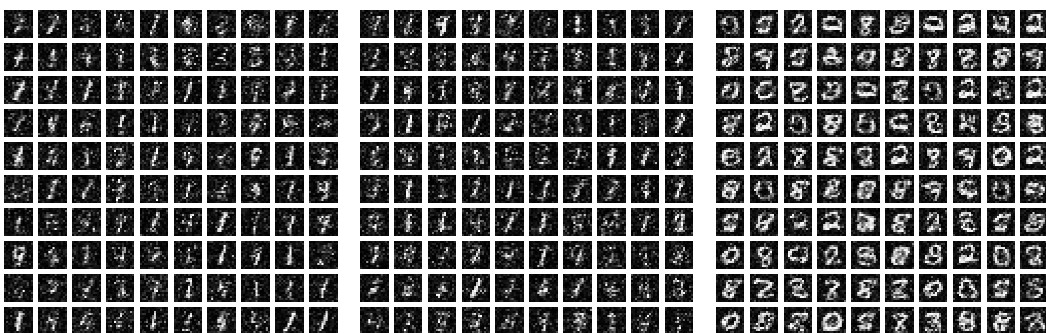

Figure 6: a) DDS , b) PIS , c) SMC. We can see how SMC mode collapses significantly as it only seems to have 5 image classes. For PIS we identified 7 classes whilst for DDS we found 9 out of 10.

as a convolutional neural network (LeCun et al., 1998) should improve the image quality over the simple feed-forward networks we have used, we believe this is the reason behind the low quality in the images as both minimising the path KL and sampling high quality images is a difficult task for a small feed-forward network.

## C.8  COMPARISON TO AIS AND MCD

We compare our results to the AIS and MCD baselines presented in Geffner & Domke (2022). Results can be seen in Tables 4 and 3, we can see that both DDS and PIS outperform LDVI across all number of steps and tasks with DDS outperforming PIS on higher values of $k$ due to inconssitent and unstable behaviour presented by PIS.

## C.9  EULER–MAYURAMA VS EXPONENTIAL INTEGRATOR

In this section we demonstrate empirically how using the Euler Mayurama integrator directly on the DDS objective leads to overestimating $\ln Z$. We compare on the Funnel dataset for which we know $\ln Z$ to be $0$ and the Ionosphere dataset which is small and for which we have a reliable estimate of $\ln Z$ via a long run SMC. Results in Table 5 show case how for lower values of $k = 64, 128$

|         | Ionosphere | | | | | | Sonar | | | | | |
|---------|------|------|------|------|------|------|------|------|------|------|------|------|
|         | ULA | MCD | UHA | LDVI | DDS | PIS | ULA | MCD | UHA | LDVI | DDS | PIS |
| $k = 64$  | -113.8 | -112.5 | -112.8 | -112.1 | -111.7 | -111.6 | -115.3 | -111.1 | -111.9 | -109.7 | -109.4 | -108.9 |
| $k = 128$ | -113.1 | -112.2 | -112.3 | -111.9 | -111.6 | -112.0 | -113.5 | -110.2 | -110.6 | -109.1 | -108.9 | -109.3 |
| $k = 256$ | -112.7 | -112.1 | -112.1 | -111.7 | -111.6 | -113.0 | -112.1 | -109.7 | -109.7 | -108.9 | -108.9 | -108.9 |

Table 3: Comparison of DDS and PIS to AIS based approaches on logistic regression models.

|  | Brownian Motion | | | | | |
|---|---|---|---|---|---|---|
|  | ULA | MCD | UHA | LDVI | DDS | PIS |
| $k = 64$ | -0.7 | 0.1 | 0.1 | - 0.5 | 1.0 | 1.0 |
| $k = 128$ | -0.3 | 0.2 | 0.4 | 0.7 | 1.1 | 1.0 |
| $k = 256$ | -0.1 | 0.5 | 0.6 | 0.9 | 1.0 | 0.7 |

Table 4: Comparison of DDS and PIS to AIS based approaches on Brownian motion time series target.

|  | Funnel | | | Ionosphere | | |
|---|---|---|---|---|---|---|
|  | Exponential Integrator | Euler Mayurama | Ground Truth | Exponential Integrator | Euler Mayurama | Ground Truth |
| $k = 64$ | $-0.206 \pm 0.059$ | $2.425 \pm 0.266$ |  | $-111.693 \pm 0.169$ | $-106.364 \pm 0.371$ |  |
| $k = 128$ | $-0.176 \pm 0.099$ | $2.011 \pm 0.532$ | 0 | $-111.587 \pm 0.136$ | $-111.158 \pm 0.399$ | $-111.560$ |
| $k = 256$ | $-0.221 \pm 0.076$ | $-0.086 \pm 0.124$ |  | $-111.575 \pm 0.163$ | $-112.271 \pm 0.366$ |  |
| $k = 512$ | $-0.176 \pm 0.068$ | $-0.154 \pm 0.066$ |  | $-111.582 \pm 0.136$ | $-112.912 \pm 0.353$ |  |

Table 5: Comparing EM vs Exponential integrators on Funnel and Ionosphere datasets. We can see how EM significantly overestimates $\ln Z$.

the Euler–Maruyama (EM) based approach significantly overestimates $\ln Z$ and overall does not perform well.

### C.9.1 DETACHED GRADIENT ABLATIONS

In this section we perform an ablation over our proposed modification of the PIS-Grad network architecture. We compare the same architecture with and without the gradient attached. We found that detaching gradient led to a favourable performance as well as more stable training. Results are presented in table 7 and were carried out for $K = 128$ and using the best found diffusion coefficient from each task.

We chose to explore this feature as it is known that optimization through unrolled computation graphs can be chaotic and introduce exploding/vanishing gradients (Parmas et al., 2018; Metz et al., 2020) which can numerically introduce bias. This has been successfully applied in related prior work (Greff et al., 2019) where detached scores are provided as features to the recognition networks. Alternative approaches exist based on smoothing the chaotic loss-landscape (Vicol et al., 2021) but we leave this for future work.

### C.10 COSINE BASED DECAY FOR PIS

Unlike DDS, with PIS it is not immediately clear how to schedule the step-sizes in a similar manner to Nichol & Dhariwal (2021). The simplest approach would be to schedule $\delta$ (the discretisation step). We perform an ablation with the cosine schedule on the discretisation step with PIS. Results are displayed in Table C.10, we can see that overall the cosine discretisation decreases or preserves the performance of PIS with the exception of the Sonar data-set in which it increases. Whilst there may be a way to improve PIS with a bespoke discretisation we were unable to find such in this work. Additionally we can see from Table C.10 how DDS improves significantly across every task when using the cosine squared schedule compared to uniform.

### C.11 PROBABILITY FLOW ODE

Figures 7 and 8 show samples obtained from the probability flow ODE of a trained DDS model. We can see that the probability flow ODE is able to perfectly sample from a uni-modal Gaussian, matters became more challenging with multi-modal distributions. Initially we discretised the ODE (12) using the same type of integrators as the SDEs to obtain $y_{k+1} = y_k + \delta\sigma^2(1 - \sqrt{1 - \alpha_{K-k}})f_\theta(K - k, y_k)$ for $y_0 \sim \mathcal{N}(0; \sigma^2 I)$. Unfortunately under this discretisation we found the probability flow ODE to become stiff with more complex distributions such as the mixture of Gaussians, resulting in strange effects in the samples (matching the modes but completely wrong shapes). By using a Heun

| Method | Funnel | LGCP | VAE | Sonar | Ionosphere | Brownian | NICE |
|---|---|---|---|---|---|---|---|
| PIS Cos decay | $-0.256 \pm 0.094$ | $501.060 \pm 0.881$ | $-110.025 \pm 0.0720$ | $-108.843 \pm 0.242$ | $-111.618 \pm 0.127$ | $1.035 \pm 0.097$ | $-4.212 \pm 0.620$ |
| PIS$_{\text{uniform}}$ | $-0.268 \pm 0.07$ | $502.002 \pm 0.957$ | $-110.025 \pm 0.070$ | $-109.349 \pm 0.356$ | $-111.602 \pm 0.145$ | $0.960 \pm 0.145$ | $-3.985 \pm 0.349$ |
| DDS | $-0.176 \pm 0.098$ | $497.944 \pm 0.994$ | $-110.012 \pm 0.071$ | $-108.903 \pm 0.226$ | $-111.587 \pm 0.136$ | $1.047 \pm 0.156$ | $-3.490 \pm 0.568$ |
| DDS$_{\text{uniform}}$ | $-0.275 \pm 0.127$ | $477.680 \pm 1.338$ | $-110.245 \pm 0.188$ | $-109.556 \pm 0.468$ | $-111.736 \pm 0.161$ | $0.475 \pm 0.166$ | $-7.047 \pm 0.691$ |

Table 6: Ablation for $\cos^2$ scheduling.

| Method | Funnel | LGCP | VAE | Sonar | Ionosphere | Brownian | NICE |
|---|---|---|---|---|---|---|---|
| PIS Grad Detach | $-0.268 \pm 0.07$ | $502.002 \pm 0.957$ | $-110.025 \pm 0.0704$ | $-109.349 \pm 0.356$ | $-111.602 \pm 0.145$ | $0.960 \pm 0.145$ | $-3.985 \pm 0.349$ |
| PIS Grad | $-0.268 \pm 0.09$ | $501.273 \pm 0.798$ | $-110.033 \pm 0.0647$ | $-109.377 \pm 0.361$ | $-111.705 \pm 0.190$ | $-5.793 \pm 0.499$ | $-3.927 \pm 0.666$ |

Table 7: Results for PIS grad with and without detaching the gradient.

integrator as proposed in Karras et al. (2022) we were able to obtain better results, however we also found we needed to increase the network size to improve the results as well as train with a learning rate decay of $0.99$. With these extra features we were able to simulate a probability flow ODE that roughly matched the marginal densities of the SDE. However even with these fixes we still found the probability flow based estimator of $\ln Z$ to drastically overestimate, this is not entirely surprising as in discrete time the expectation of this estimator is not guaranteed to be an ELBO.

### C.12 UNDERDAMPED OU RESULTS

We perform some additional experiments with the underdamped OU reference process. Similar to the damped setting we parametrise the network as:

$$f_\theta(k, x, p) = \text{NN}_1(k, x, p; \theta) + \text{NN}_2(k; \theta)\nabla \ln \pi(x). \tag{74}$$

Unfortunately, unlike in DDS and PIS we cannot directly aid the update/proposal for $x_t$ with $\nabla \ln \pi(x)$, thus this naive parametrisation is not the ideal inductive bias.

Results in Figures 9 and 10 show some experiments for this approach. The results perform worse than DDS for a standard OU reference process and PIS. However they are still better than VI. We believe that future work exploring better inductive biases for the network $f_\theta(k, x, p)$ could narrow the performance gap in this approach. Additionally we highlight that in Figure 9 (Funnel Distribution) we can see the underdamped approach overestimates $\ln Z$ for $k = 64$. We believe this may be due to numerical error when generating a sample.

### C.13 BOX PLOTS FOR MAIN RESULTS

For completeness in this section we also present our results via box plots as can be seen in Figures 11 and 12.

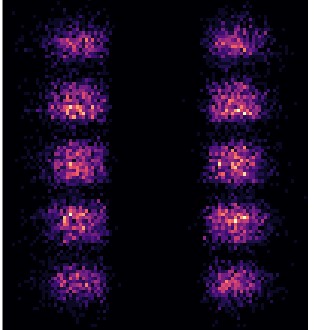

Figure 7: a) Target distribution (MoG), b) DDS SDE samples, c) trained probability flow ODE.

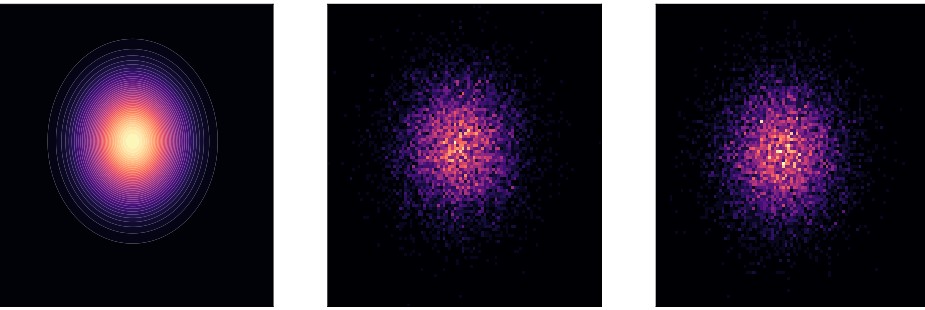

Figure 8: a) Target distribution $\mathcal{N}(6,1)$, b) DDS SDE c) Trained DDS probability flow ODE.

# D DENOISING DIFFUSION SAMPLERS IN DISCRETE-TIME: FURTHER DETAILS

## D.1 FORWARD PROCESS AND ITS TIME REVERSAL

In discrete-time, we consider the following discrete-time "forward" Markov process

$$p(x_{0:K}) = \pi(x_0) \prod_{k=1}^{K} p_{k|k-1}(x_k|x_{k-1}). \tag{75}$$

Following (17), we use $p_{k|k-1}(x_k|x_{k-1}) = \mathcal{N}(x_k; \sqrt{1-\alpha_k}x_{k-1}, \sigma^2\alpha_k I)$. The parameters $(\alpha_k)_{k=1}^K$ are such that $p_K(x_K) \approx \mathcal{N}(x_K; 0, \sigma^2 I)$.

We propose here to sample approximately from $\pi$ by approximating the ancestral sampling scheme for (75) corresponding to the backward decomposition

$$p(x_{0:K}) = p_K(x_K) \prod_{k=1}^{K} p_{k-1|k}(x_{k-1}|x_k), \text{ with } p_{k-1|k}(x_{k-1}|x_k) = \frac{p_{k-1}(x_{k-1})p_{k|k-1}(x_k|x_{k-1})}{p_k(x_k)}, \tag{76}$$

where $p_0(x_0) = \pi(x_0)$. If we could thus sample $x_K \sim p_K(\cdot)$ then $x_{k-1} \sim p_{k-1|k}(\cdot|x_k)$ for $k = K, ..., 1$ then $x_0$ would indeed be a sample from $\pi$. However as the marginal densities $(p_k)_{k=1}^K$ are not available in closed-form, we cannot implement exactly this ancestral sampling procedure.

As we have by design $p_K(x_K) \approx \mathcal{N}(x_K; 0, \sigma^2 I)$, we can simply initialize the ancestral sampling procedure by a Gaussian sample. The approximation of the backward Markov kernels $p_{k-1|k}(x_{k-1}|x_k)$ is however more involved.

## D.2 REFERENCE PROCESS AND BELLMAN RECURSION

We introduce the "simple" reference process $p^{\text{ref}}(x_{0:K})$ defined by

$$p^{\text{ref}}(x_{0:K}) = \mathcal{N}(x_0; 0, \sigma^2 I)p(x_{1:K}|x_0) = \mathcal{N}(x_0; 0, \sigma^2 I) \prod_{k=1}^{K} p_{k|k-1}(x_k|x_{k-1}) \tag{77}$$

which is designed by construction to admits the marginal distributions $p_k^{\text{ref}}(x_k) = \mathcal{N}(x_k; 0, \sigma^2 I)$ for all $k$. It can be easily verified using the chain rule for KL that the extended target process $p$ is the distribution minimizing the reverse (or forward) KL discrepancy w.r.t. $p^{\text{ref}}$ over the set of path measures $q(x_{0:K})$ with marginal $q_0(x_0) = \pi(x_0)$ at the initial time, i.e.

$$p = \arg\min_q \left\{ \text{KL}(q||p^{\text{ref}}) : q_0 = \pi \right\}.$$

The intractable backward Markov densities $p_{k-1|k}$ one wants to approximate can also be rewritten as twisted versions of the tractable backward Markov densities $p_{k-1|k}^{\text{ref}}$ by some value functions $(\phi_k)_{k=0}^K$.

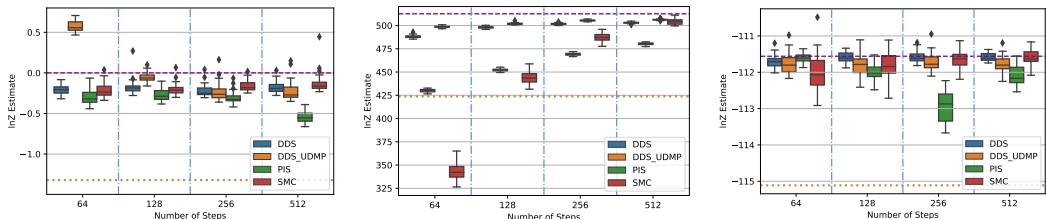

Figure 9: $\ln Z$ estimate as a function of number of steps $K$ - a) Funnel , b) LGCP, c) Logistic Ionosphere dataset. Yellow Dotted line is MF-VI and dashed magenta is the gold standard.

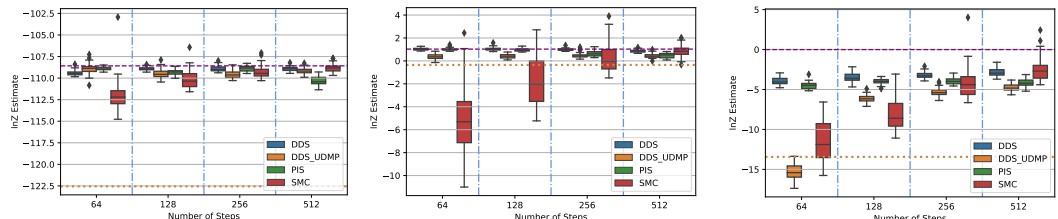

Figure 10: $\ln Z$ estimate as a function of number of steps $K$ - a) Logistic Sonar dataset, b) Brownian motion, c) NICE. Yellow dotted line is MF-VI and dashed magenta is the gold standard.

**Proposition 7.** *We have for $k = 1, ..., K$*

$$p_{k-1|k}(x_{k-1}|x_k) = \frac{\phi_{k-1}(x_{k-1})p^{\mathrm{ref}}_{k-1|k}(x_{k-1}|x_k)}{\phi_k(x_k)}, \quad for \quad \phi_k(x_k) = \frac{p_k(x_k)}{p^{\mathrm{ref}}_k(x_k)}. \tag{78}$$

*The value functions $(\phi_k)_{k=1}^K$ satisfy a forward Bellman type equation*

$$\phi_k(x_k) = \int \phi_{k-1}(x_{k-1})p^{\mathrm{ref}}_{k-1|k}(x_{k-1}|x_k)\mathrm{d}x_{k-1}, \qquad \phi_0(x_0) = \frac{\pi(x_0)}{p^{\mathrm{ref}}_0(x_0)}. \tag{79}$$

*It follows that*

$$\phi_k(x_k) = \int \phi_0(x_0) \ p^{\mathrm{ref}}_{0|k}(x_0|x_k)\mathrm{d}x_0. \tag{80}$$

*Proof.* To establish (78), we use Bayes' rule

$$\begin{aligned} p_{k-1|k}(x_{k-1}|x_k) &= \frac{p_{k-1}(x_{k-1})p_{k|k-1}(x_k|x_{k-1})}{p_k(x_k)} \\ &= \frac{\phi_{k-1}(x_{k-1})p^{\mathrm{ref}}_{k-1}(x_{k-1})p_{k|k-1}(x_k|x_{k-1})}{\phi_k(x_k)p^{\mathrm{ref}}_k(x_{k-1})} \\ &= \frac{\phi_{k-1}(x_{k-1})p^{\mathrm{ref}}_{k-1|k}(x_{k-1}|x_k)}{\phi_k(x_k)}, \end{aligned}$$

where we have used the fact that $p_k(x_k) = \phi_k(x_k)p^{\mathrm{ref}}_k(x_k)$, $p^{\mathrm{ref}}_{k|k-1}(x_k|x_{k-1}) = p_{k|k-1}(x_k|x_{k-1})$ and Bayes' rule again. Now we have $\int p_{k-1|k}(x_{k-1}|x_k)\mathrm{d}x_{k-1} = 1$ for any $x_k$, so it follows directly from the expression of this transition kernel that the value function satisfies

$$\phi_k(x_k) = \int \phi_{k-1}(x_{k-1})p^{\mathrm{ref}}_{k-1|k}(x_{k-1}|x_k)\mathrm{d}x_{k-1}, \qquad \phi_0(x_0) = \frac{\pi(x_0)}{p^{\mathrm{ref}}_0(x_0)}. \tag{81}$$

By iterating this recursion, it follows that

$$\phi_k(x_k) = \int \phi_0(x_0) \ p^{\mathrm{ref}}_{0|k}(x_0|x_k)\mathrm{d}x_0. \tag{82}$$

$\square$

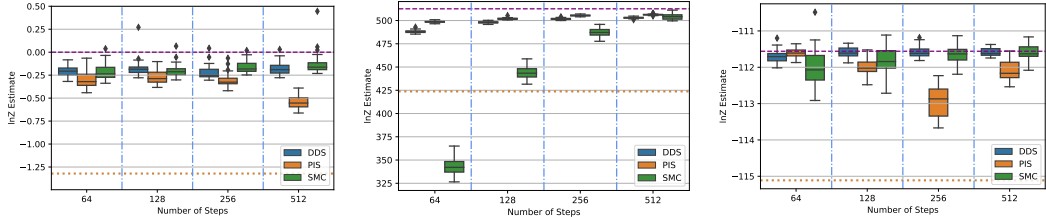

Figure 11: $\ln Z$ estimate as a function of number of steps $K$ - a) Funnel , b) LGCP, c) Logistic Ionosphere dataset. Yellow Dotted line is MF-VI and dashed magenta is the gold standard.

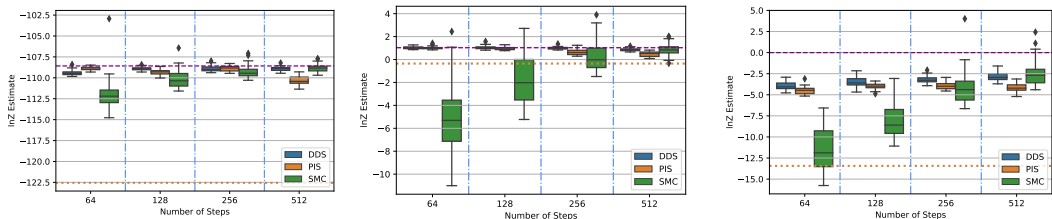

Figure 12: $\ln Z$ estimate as a function of number of steps $K$ - a) Logistic Sonar dataset, b) Brownian motion, c) NICE. Yellow dotted line is MF-VI and dashed magenta is the gold standard.

### D.3 APPROXIMATING THE BACKWARD KERNELS

We want to approximate the backward Markov transitions $p_{k-1|k}$. From (78), this would require not only to approximate the value functions - Monte Carlo estimates based on (80) are high variance - but also to sample from the resulting twisted kernel which is difficult. However, if we select $\beta_k \approx 0$, then we have $\phi_{k-1}(x) \approx \phi_k(x)$ and by a Taylor expansion of $\nabla \ln \phi_k$ around $x_k$ we obtain

$$p_{k-1|k}(x_{k-1}|x_k) = p^{\text{ref}}_{k-1|k}(x_{k-1}|x_k) \exp(\ln \phi_{k-1}(x_{k-1}) - \ln \phi_k(x_k))$$
$$\approx \mathcal{N}(x_{k-1}; \sqrt{1-\alpha_k}x_k + \sigma^2(1-\sqrt{1-\alpha_k})\nabla \ln \phi_k(x_k)). \tag{83}$$

By differentiating the identity (80) w.r.t. $x_k$, we obtain the following expression for $\phi_k$

$$\nabla \ln \phi_k(x_k) = \int \nabla \ln p^{\text{ref}}_{0|k}(x_0|x_k) \; p_{0|k}(x_0|x_k)\mathrm{d}x_0. \tag{84}$$

Alternatively, we can also use $\nabla \ln \phi_k(x_k) = \nabla \ln p_k(x_k) + \frac{x}{\sigma^2}$ where $\nabla \ln p_k(x_k)$ can be easily shown to satisfy

$$\nabla \ln p_k(x_k) = \int \nabla \ln p_{k|0}(x_k|x_0) \; p_{0|k}(x_0|x_k)\mathrm{d}x_0. \tag{85}$$

For DDPM, the conditional expectation in (85) is reformulated as the solution to a regression problem, leveraging the fact that we can easily obtain samples from $\pi(x_0)p_{k|0}(x_k|x_0)$ as $\pi$ is the data distribution in this context. In the Monte Carlo sampling context considered here, we could sample approximately from $p_{0|k}(x_0|x_k) \propto \pi(x_0)p_{k|0}(x_k|x_0)$ using MCMC so as to approximate the conditional expectations in (84) and (85) but this would defeat the purpose of the proposed methodology. We propose instead to approximate $\nabla \ln \phi_k$ by minimizing a suitable criterion.

In practice, we will consider a distribution $q^\theta(x_{0:K})$ approximating $p(x_{0:K})$ of the form

$$q^\theta(x_{0:K}) = \mathcal{N}(x_K; 0, \sigma^2 I) \prod_{k=1}^K q^\theta_{k-1|k}(x_{k-1}|x_k), \tag{86}$$

i.e. we select $q^\theta_K(x_K) = q_K(x_K) = \mathcal{N}(x_K; 0, \sigma^2 I)$ as $p_K(x_K) \approx \mathcal{N}(x_K; 0, \sigma^2 I)$. We want $q^\theta_{k-1|k}(x_{k-1}|x_k)$ to approximate $p_{k-1|k}(x_{k-1}|x_k)$, i.e. $f_\theta(k, x_k) \approx \nabla \ln \phi_k(x_k)$. Inspired by (83), we will consider an approximation of the form

$$q^\theta_{k-1|k}(x_{k-1}|x_k) = \mathcal{N}(x_{k-1}; \sqrt{1-\alpha_k}x_k + \sigma^2(1-\sqrt{1-\alpha_k})f_\theta(k, x_k), \sigma^2 \alpha_k I). \tag{87}$$

where $f_\theta(k, x_k)$ is differentiable w.r.t. $\theta$.

### D.4 Hyperparameters

#### D.4.1 Fitted Hyperparameters

To aid reproducibility we report all fitted hyperparameters for each of our methods and PIS across all experiments in Tables 8 and 9.

#### D.4.2 Optimisation Hyperparameters

As mentioned in the experimental section across all experiments modulo the Funnel we use the Adam optimiser with a learning rate of $0.0001$ with no learning decay and $11000$ training iterations, for the rest of the optimisation parameters use the default settings as provided by the Optax library (Hessel et al., 2020) which are $b_1 = 0.9, b_2 = 0.999, \epsilon = 10^{-8}$ naming as per Kingma & Ba (2015).

From the github repository of Zhang & Chen (2022) we were only able to find hyperparameters reported for the Funnel distribution. In order to first reproduce their results we used the a learning rate of $0.005$ and a learning rate decay of $0.95$ as per their implementation, their results were initially not reproducible due to a bug in setting $\sigma_f = 1$ despite comparing to methods at the less favourable values of $\sigma_f = 3$. For $\sigma_f = 1$ we were able to reproduce their results. However we report results at $\sigma_f = 3$ as this is the traditional value used for this loss. As no other optimisation configuration files were reported we used the more conservative learning rate of $0.0001$ since PIS was very unstable for $0.005$ with decay $0.95$ across many of our tasks. Finally we would like to clarify that the exact same optimiser settings where used for both PIS and DDS in order to ensure a fair comparison.

#### D.4.3 Drift and Gradient Clipping

We follow the same gradient clipping as in Zhang & Chen (2022) that is :

$$f_\theta(k, x) = \text{clip}\Big(\text{NN}_1(k, x; \theta) + \text{NN}_2(k; \theta) \odot \text{clip}\big(\nabla \ln \pi(x), -10^2, 10^2\big), -10^4, 10^4\Big) \tag{88}$$

| | funnel | | | lgcp | | | ion | | |
|---|---|---|---|---|---|---|---|---|---|
| | DDS | PIS | UDMP | DDS | PIS | UDMP | DDS | PIS | UDMP |
| $K=64$ | $\sigma=1.075,\alpha=1.075$ | $\sigma=1.068$ | $\sigma=1.85,\alpha=1.67,m=0.9$ | $\sigma=2.1,\alpha=1.50$ | $\sigma=1.068$ | $\sigma=1.1,\alpha=2.5,m=0.4$ | $\sigma=0.688,\alpha=1.463$ | $\sigma=0.253$ | $\sigma=0.6,\alpha=3.85,m=0.600$ |
| $K=128$ | $\sigma=1.075,\alpha=0.6875$ | $\sigma=0.416$ | $\sigma=1.85,\alpha=3.7,m=0.9$ | $\sigma=2.1,\alpha=0.75$ | $\sigma=0.742$ | $\sigma=1.4,\alpha=2.5,m=0.4$ | $\sigma=0.3,\alpha=1.075$ | $\sigma=0.09$ | $\sigma=0.6,\alpha=3.85,m=0.600$ |
| $K=256$ | $\sigma=1.85,\alpha=0.3$ | $\sigma=0.742$ | $\sigma=1.075,\alpha=2.5,m=0.9$ | $\sigma=2.1,\alpha=0.900$ | $\sigma=0.579$ | $\sigma=1.4,\alpha=4.5,m=0.4$ | $\sigma=0.3,\alpha=0.688$ | $\sigma=0.416$ | $\sigma=0.6,\alpha=3.85,m=1.0$ |
| $K=512$ | $\sigma=1.463,\alpha=0.3$ | $\sigma=0.253$ | $\sigma=0.688,\alpha=3.7,m=0.9$ | $\sigma=2.1,\alpha=1.500$ | $\sigma=0.416$ | $\sigma=1.7,\alpha=4.5,m=0.4$ | $\sigma=0.688,\alpha=0.688$ | $\sigma=0.253$ | $\sigma=0.6,\alpha=3.85,m=1.0$ |

Table 8: Fitted hyperparameters for funnel, lgcp and ion experiments.

| | lr_sonar | | | vae | | | brownian | | | nice | | |
|---|---|---|---|---|---|---|---|---|---|---|---|---|
| | DDS | PIS | UDMP | DDS | PIS | UDMP | DDS | PIS | UDMP | DDS | PIS | UDMP |
| $K=64$ | $\sigma=0.3,\alpha=1.650$ | $\sigma=0.253$ | $\sigma=1.15,\alpha=1.7,m=2.2$ | $\sigma=0.61,\alpha=2.2$ | $\sigma=0.253$ | NA | $\sigma=0.1,\alpha=2.35$ | $\sigma=0.084$ | $\sigma=0.115,\alpha=4.8,m=2.2$ | $\sigma=1.5,\alpha=2.125$ | $\sigma=0.75$ | $\sigma=1.2,\alpha=1.0,m=0.9$ |
| $K=128$ | $\sigma=0.3,\alpha=1.2$ | $\sigma=0.253$ | $\sigma=0.55,\alpha=1.7,m=3.1$ | $\sigma0.61,\alpha=1.670$ | $\sigma=0.2523$ | NA | $\sigma=0.1,\alpha=1.8$ | $\sigma=0.093$ | $\sigma=0.115,\alpha=4.8,m=2.2$ | $\sigma=1.5,\alpha=1.75$ | $\sigma=0.588$ | $\sigma=1.2,\alpha=1.0m=1.65$ |
| $K=256$ | $\sigma=0.3,\alpha=0.75$ | $\sigma=0.253$ | $\sigma=0.55,\alpha=2.9,m=2.2$ | $\sigma=0.61,\alpha=1.140$ | $\sigma=0.506$ | NA | $\sigma=0.1,\alpha=2.35$ | $\sigma=0.043$ | $\sigma=0.115,\alpha=3.75,m=2.2$ | $\sigma=1.5,\alpha=1.75$ | $\sigma=0.425$ | $\sigma=1.2,\alpha=2.5,m=0.9$ |
| $K=512$ | $\sigma=0.3,\alpha=0.75$ | $\sigma=0.253$ | $\sigma=0.55,\alpha=2.9,m=3.1$ | $\sigma=0.61,\alpha=1.670$ | $\sigma=0.416$ | NA | $\sigma=0.1,\alpha=1.8$ | $\sigma=0.0408$ | $\sigma=0.115,\alpha=4.8,m=2.2$ | $\sigma=1.5,\alpha=2.5$ | $\sigma=0.263$ | $\sigma=1.2,\alpha=2.5,m=1.65$ |

Table 9: Fitted hyperparameters for sonar, brownian, vae and nice experiments.

