# OpenReview forum: "Denoising Diffusion Samplers"
_ICLR.cc/2023/Conference — ICLR 2023 poster_

### Official Review · Reviewer_cpvn · 2022-10-21

**Confidence:** 1
**Correctness:** 4
**Technical Novelty And Significance:** 3
**Empirical Novelty And Significance:** 2
**Recommendation:** 5

**Clarity, Quality, Novelty And Reproducibility:**

Super difficult to follow for everyone that is not an expert in both diffusion models and MCMC. It is hard to me to assess originality. I bet the authors are not the first ones to attempt some connection between diffusion models and MCMC, but what is and what is not a contribution is not really discussed in the paper, and I am not knowledgeable enough to guess it myself.

**Strength And Weaknesses:**

The paper is definitely timely, considering the hype around diffusion models, and it reads pretty rigorous. I guess it may be of interest to the very few readers able to understand it.
On the downside, I must say that I had great difficulty following it, because my math is not strong enough. As a practitionner, I am left with some frustrating feeling that I probably don't really even know how to use that sampler in practice, the whole thing being quite burried in the maths.
My guess is: this paper is very nice for a strong theoretical connection between diffusion models and samplers, but I can't really assess it.

Some attempts of comments on the go:
* below (2), I guess you mean \lambda_T, not \alpha_T
* maybe it is trivial, but I probably missed a proof for what is claimed in the first paragraph of 2.2
* between (5) and (6), isn't there a minus sign "-" mismatch ?
* "where we have use" -> "used"
* It looks like only the equation below (8) $f_\theta(t,x) = ....$ is necessary, not really (8), which looks like just a copy of (5)
* the discussion in 2.4 and what it means as a difference with the section before is difficult for me to understand
* above (14), I don't know what \pi(x; 0, \sigma^2 I) means. This section looks like some extension of section 2, and I must confess I just gave up following the overwhelming amount of maths at this point.


**Summary Of The Paper:**

The paper describes sampling (MCMC parlance) as a diffusion problem. In the forward direction, the target distribution diffuses to Gaussian noise, while the reverse direction of interest allows diffusing from noise to the target distribution. As can be seen and is highlighted by the authors, this problem is classical in MCMC.
They develop a sampler which is inspired by recent advances in diffusion based models. Building a reference diffusion process that diffuses from noise to noise, they describe some particular criterion they optimize to train their sampler (eq 10), basically based on minimizing the KL divergence between the distribution it yields and the reference distribution.
The same story is then told again for different kinds of flavours, involving  augmenting the original state by a momentum variables and also studying the discrete case, which is of interest for implementations.
Experiments are numerous and mostly done on small data, but apparently matching what is usually done in the field.

**Summary Of The Review:**

Probably correct theoretical paper about the connections between diffusion models and MCMC. Hard to follow. It doesn't look like great care was taken to help the practitionner use the approach.

---

> ### Author Response · Authors · 2022-11-18
> **Rebuttal and thank you for your review !**
>
> We would like to thank the reviewer for taking the time to assess our work and provide very useful feedback overall strengthening the presentation of our manuscript. We apologize for the nature of the mathematical presentation of our work and hope to clarify the contributions of our approach in the remainder of this response.
>
> > "... doesn't look like great care was taken to help the practitioner use the approach."
>
> We have added pseudocode for the approach in Algorithm 1. Furthermore we have added tables 7 and 8 to the appendix which contain all used hyperparameters across experiments. This should allow practitioners to use the approach and hopefully explore/utilize DDS. Finally we have added a new section on the integrator/discretisation of $\mathcal{P}^{\mathrm{ref}}$ which is what motivates our final algorithm and is particularly relevant to practitioners as it details the conditions required when implementing/discretizing these processes in order to have a valid ELBO for ln Z.
>
> Finally in addition to the novelty of our proposed approach we empirically demonstrate how it is significantly easier to train (more numerically stable) than previous neural SDE approaches for sampling such as PIS. We hope it should be of particular interest and use to the practitioner. We have also added a simple analytic Gaussian target example in Appendix A.2 which motivates conceptually why DDS is more numerically stable than PIS.
>
> > "maybe it is trivial, but I probably missed a proof for what is claimed in the first paragraph of 2.2"
>
> The proof sketch for proposition 1 in Appendix A.1 illustrates how the statements in the discussion of the 2.2 paragraph arise. The steps in this sketch illustrate various ways in which the distribution of $\mathcal{P}$ can be decomposed in terms of $\mathcal{P}^{\mathrm{ref}}$ and the marginals of $\mathcal{P}$ and $\mathcal{P}^{\mathrm{ref}}$ respectively. From these decompositions we can arrive at the KL based formulations of time reversal that are discussed in first paragraph of 2.2. and proposition 1. These steps are at the core of why we need an exponential integrator and can also be seen in the proof for proposition 2.
>
> > "I bet the authors are not the first ones to attempt some connection between diffusion models and MCMC, but what is and what is not a contribution is not really discussed in the paper."
>
> To the best of our knowledge and at the time of writing we are not aware of any paper applying denoising diffusion based models to black-box sampling and inference as well as establishing the relevant connections required. The most related work being the PIS approach does not allude to or establish any connection to denoising diffusion models. To clarify some of the major contributions of this work are:
>
> * We port over denoising diffusion based modeling from generative modeling to sampling
> * Such transition was not simple as the standard score/regression based objectives used in DDPM do not directly apply to the sampling setting since we do not have access to samples of the process diffusing the target distribution.
> * There are very careful design choices that must be made when using an OU process in conjunction with the objectives we consider. One that is particularly important for the practitioner is that when discretizing the continuous time objective one must use an integrator that preserves the marginals of reference SDE (in this case the OU process) otherwise the expectation of the resulting ln Z estimator is not an ELBO. This is of crucial importance for empirical evaluation as many approaches compare different inference/sampling schemes in practice by showcasing which one has a higher expectation of the ln Z estimate, if we are not guaranteed to have a lower bound then said comparisons cannot be done reliably. We have added proposition 2 that formalizes this as well as including some empirical evaluation in Table 5 that showcases this issue. Moving forward this constitutes of a useful case study than can inform future methods based on similar objectives.
> * We demonstrate empirically that our approach is stabler/easier to train than prior work using neural SDEs as samplers (PIS).
> * We derive novel estimators for ln Z based on normalizing flows that do not apply to prior work. Whilst the empirical performance of these was not beneficial in this work we hope it has the potential to inform future DDS based approaches to sampling.
>
> Finally the reviewer highlighted several typos which have been corrected in the latest version of the manuscript, and to improve readability we have moved some of the details of the underdamped approach to the appendix focusing more on the novelty and strict requirement of the expoential integrator in the context of inference for DDS. As before we would like to thank the reviewer for taking extra care in spotting these, we hope these comments / ammendments address the reviewers concerns.

---

> ### Author Response · Authors · 2022-12-06
> **Reminder - Phase 2 discussion period coming to an end (12 of December)**
>
> Dear Reviewer,
>
> This is a short reminder that the phase 2 dicussion period is coming to an end on the 12th of December. As we have modified the structure of the main manuscript adding new theoretical and experimental results to improve the presentation and motivation of the work we believe that we have hopefully addressed most of your comments (in conjunction with the rebuttal). We hope that the reviewer can re-asses the new improved manuscript and we are happy to address any additional questions or comments that may arise.
>
> As before we thank the reviewer for their initial review and helpful feedback.

---

### Official Review · Reviewer_SEx8 · 2022-10-24

**Confidence:** 3
**Correctness:** 3
**Technical Novelty And Significance:** 3
**Empirical Novelty And Significance:** 3
**Recommendation:** 6

**Clarity, Quality, Novelty And Reproducibility:**

The method is novel both from practical and theoretical standpoint. The algorithm should be easy to reproduce but the code is not provided.

**Strength And Weaknesses:**

Strength: novel theoretical approach to sampling from non-normalised density

Weakness: absent experimental comparison with Langevin Monte-Carlo methods as they aim to solve the same problem.

**Summary Of The Paper:**

Authors propose novel theoretical framework based on Girsanov formula to learn sampler based on reversing diffusion process from non-normalised density function.

**Summary Of The Review:**

The proposed approach definitely interesting and I am eager to see whether it can be successfully applied for non-convex optimization. I believe that this result is worth publishing due to novel approach for sampling.

---

> ### Author Response · Authors · 2022-11-18
> **Rebuttal and thank you for your review !**
>
>
> We would like to thank the reviewer for taking the time to read our work and for the encouraging comments. We will now proceed to address the highlighted weaknesses.
>
> > "Absent experimental comparison with Langevin Monte-Carlo methods as they aim to solve the same problem."
>
> We have compared to Langevin Monte Carlo, HMC, and modern hybrid VI/Langevin based approaches such as MCD and LDVI. These comparisons can be found in Tables 2 and 4 Appendix C.3. Adding an ascii version of the table for ease of discussion:
>
> |         | Ionosphere |        |        |        |        |        |  Sonar |        |        |        |        |        |
> |---------|:----------:|:------:|:------:|:------:|:------:|:------:|:------:|:------:|:------:|:------:|:------:|:------:|
> |         | ULA        |   MCD  |   UHA  |  LDVI  |   DDS  |   PIS  |   ULA  |   MCD  |   UHA  |  LDVI  |   DDS  |   PIS  |
> | $k=64$  | -113.8     | -112.5 | -112.8 | -112.1 | -111.7 | -111.6 | -115.3 | -111.1 | -111.9 | -109.7 | -109.4 | -108.9 |
> | $k=128$ | -113.1     | -112.2 | -112.3 | -111.9 | -111.6 | -112.0 | -113.5 | -110.2 | -110.6 | -109.1 | -108.9 | -109.3 |
> | $k=256$ | -112.7     | -112.1 | -112.1 | -111.7 | -111.6 | -113.0 | -112.1 | -109.7 | -109.7 | -108.9 | -108.9 | -108.9 |
>
>
> |         | Brownian Motion |     |     |       |     |     |
> |---------|:---------------:|-----|-----|-------|-----|-----|
> |         | ULA             | MCD | UHA | LDVI  | DDS | PIS |
> | $k=64$  | -0.7            | 0.1 | 0.1 | - 0.5 | 1.0 | 1.0 |
> | $k=128$ | -0.3            | 0.2 | 0.4 | 0.7   | 1.1 | 1.0 |
> | $k=256$ | -0.1            | 0.5 | 0.6 | 0.9   | 1.0 | 0.7 |
>
> We can see that DDS (our approach) outperforms ULA (Langevin Monte Carlo based estimator) , UHA (HMC based estimator) and MCD, across all numbers of steps. Furthermore we outperform LDVI for a smaller number of steps.
>
> > "The algorithm should be easy to reproduce but the code is not provided."
>
> We have included hyperparameters (for DDS , PIS and underdampened) for all of our experiments in order to make it easy to reproduce our results.
>
> As we have addressed the main weakness in this review and provide comparisons to Langevin based Monte Carlo among other MCMC based approaches, we hope that the reviewer can update their review/score to address the changes.

---

> ### Author Response · Authors · 2022-12-06
> **Reminder - Phase 2 discusion period coming to an end (12 of December)**
>
> Dear Reviewer,
>
> This is a short reminder that the phase 2 dicussion period is coming to an end on the 12th of December. As we have addressed your main weakness and provided comparisons to Langevin dynamics as well as HMC baselines we hope that the reviewer can re-asses/update their review. Additionally we have strenghtened the manuscript overall addming many other empirical and theoretical results that strenthen the motivation of our approach.
>
> Finally we are happy to address any outstanding remarks and thank you again for your review.

---

> > ### Comment · Reviewer_SEx8 · 2022-12-12
> > **Thank you**
> >
> > Thank you for your response, new added results and comparison with Langevin really important additions.
> >
> > Though I had not much time to review new results/improvement, I’m definitely more confident that paper is worth publishing.

---

> > > ### Author Response · Authors · 2022-12-12
> > > **Thank you !**
> > >
> > > Dear Reviewer,
> > >
> > > Thank you for the very prompt response and updated confidence.

---

### Official Review · Reviewer_bg9B · 2022-10-25

**Confidence:** 3
**Correctness:** 3
**Technical Novelty And Significance:** 3
**Empirical Novelty And Significance:** 3
**Recommendation:** 6

**Clarity, Quality, Novelty And Reproducibility:**

The paper is relatively clear except sections on Underdamped Ornstein-Uhlenbeck which are hard to read due to an overabundance of in-line formulae. The figures are hard to see. The figures should also be referenced in the text, as should the tables. Empirical evaluation could see some improvements. The approach seems novel, although heavily inspired by DDPM.


**Strength And Weaknesses:**

Strength: the method is connected to Denoising Diffusion generative models, which gained much popularity.

Weakness: I have concerns regarding the empirical evaluation and the clarity of the presentation. The figures which are supposed to represent comparison against other sampling methods (mainly SMC and path integral sampler) are barely discernible. But it can be seen that the DDS seems to work on par with PIS. It is also not clear the training times that are mentioned to be non-negligible compared to SMC. Besides, the experiments seem to be relatively small or medium scale.

Regarding the style of the presentation it is sometimes obscured by heavy-weight formulae, e.g. in section 3.2. I think proper formatting should remedy that. Tables do not seem to be referenced anywhere, and Figures 2, 3 neither. Sections' 3.1, 3.2 titles contain the same typo: "Ulhenbeck".

**Summary Of The Paper:**

The paper introduces Denoising Diffusion Sampler. The task at hand is to sample from an un-normalized target density with an unknown normalization constant; and also to estimate said constant. Inspired by Denoising Diffusion generative models, the paper proposes a reverse-time SDE of a process that diffuses the target distribution to the Gaussian distribution. The authors propose an approximate learning objective so that to get rid of intractable scores of the marginals of the process. The authors evaluate their method on benchmarks, comparing it to other sampling methods.

**Summary Of The Review:**

My issue with the paper lies within its empirical evaluation and comparative advantage of the proposed method as it relates to other sampling techniques; and also within the clarity of the presentation.

UPDATE: after the discussion period, I am willing to increase the score.

---

> ### Author Response · Authors · 2022-11-18
> **Rebuttal and thank you for your review !**
>
> We would like to thank the reviewer for the detailed and insightful review. We have incorporated all the typographical/notational suggestions as well as made amendments to figures/plots. We will now go over the main criticisms and detail how they have been addressed.
>
> > "The figures which are supposed to represent comparison against other sampling methods (mainly SMC and path integral sampler) are barely discernible."
>
> We have replaced the box plots with line plots in the revised version. We agree with the reviewer that the previous box plots were difficult to read and overly cluttered. We find that the line plots exhibit the difference in performance across methods much more clearly.
> But it can be seen that the DDS seems to work on par with PIS.
>
> In the new plots it is clear DDS outperforms PIS across all tasks with the exception of the LGCP target, where we can see that both DDS and SMC are mixing slowly due to the fact that the components of the LGCP target live on very different scales, some are very large in particular. Furthermore we can see that PIS becomes unstable as K is increased, ultimately degrading its performance across many tasks where DDS significantly outperforms it (see Figure 1). PIS training dynamics are significantly more unstable and do not stick the landing (i.e. stabilize when the minima is reached) in comparison to DDS. Overall our empirical evaluation suggests that DDS is a more reliable method with better numerical properties as well as overall performance. To further back this up we have added a new section A.2 which analytically computes the drifts for DDS and PIS for a Gaussian target. Corollaries 1,2 show that for a target $\mathcal{N}(\alpha, I)$ the magnitude for the DDS drift time $t=0$  is scaled by $O(\exp(-T))$, whilst for PIS it is $O(1/T)$ illustrating how DDS will have a significantly smaller drift magnitude thus making it more stable.
>
> > "Besides, the experiments seem to be relatively small or medium scale. … Empirical evaluation could see some improvements"
>
> We would like to highlight that we explore the standard benchmarks typically presented in inference/sampling papers (Funnel, LGCP, Ionosphere , SONAR, and Brownian) in addition to this we also include some neural network based benchmarks such as NICE and VAE.
>  Furthermore we have several ablations comparing the different features (e.g. integrator, gradient detach) proposed in this work. Finally for MCMC based methods we would like to highlight that targets of dimension 1600 (LGCP) and 196 (NICE) are by no means regarded small scale. Overall the average dimension across experiments ~ 280.14 which again is not small scale in the MCMC world.
>
> > "figures should also be referenced in the text, as should the tables"
>
> We thank the reviewer for highlighting this and have amended it in the revised version of the manuscript.
>
> > "The approach seems novel, although heavily inspired by DDPM."
>
> As clearly acknowledged in our paper, the high level idea in our approach is the same as DDPM. However, the final objective and method are very different. The objective we derive is more in line with path integral control based objectives. However as we demonstrate both formally and empirically (Proposition 2 and Table 5) when discretizing these continuous-time objectives one has to carefully design the integrator in order to obtain a valid ELBO (and avoid overestimating ln Z). Overall this leads to an algorithm significantly different from DDPM or PIS.
>
> Overall we thank the reviewer for their thankful comments in this response we have highlighted the relevant scale of our experimentation, improved figures as well as presentation and motivation (e.g. the new section 3.1. on the need for the exp integrator), we have also pointed out additional comparisons and results explored in the appendix. Furthermore we highlight the increased training stability in our approach which has a clear increase in performance across most tasks we explore. Finally we have moved  some of the less relevant underdamped details to the appendix where they are explained more thoroughly in order to focus the reader and improve the overal presentation. We hope that these changes address the reviewers concerns and that they can re-evaluate the improved version of our manuscript.

---

> > ### Comment · Reviewer_bg9B · 2022-12-12
> > **Reply to the authors**
> >
> > Thank you for your reply and the hard work put into making the paper better! After reading the discussion, I would like to increase my score.

---

> > > ### Author Response · Authors · 2022-12-12
> > > **Thank you !**
> > >
> > > Dear Reviewer,
> > >
> > > Thank you for the very prompt response and updated score. As before it is thank to all the reviews that have contributed making this paper better so thank you again.

---

> ### Author Response · Authors · 2022-12-06
> **Reminder - Phase 2 discussion period coming to an end soon (12 of December)**
>
> Dear Reviewer,
>
> This is a short reminder that the phase 2 dicussion period is coming to an end on the 12th of December. As we have provided remarks addressing every question the reviewer has asked as well as made significant changes to the manuscript (both in presentation as well as new results) we hope that the reviewer can re-asses our work. As reviewer Vb4W has highlighted the revised version has improved significantly in value, thus we hope in conjunciton with the rebuttal it has also addressed your concerns and suggestions.
>
> Furthermore we are happy to answer/address any outstanding remarks for the reminder of this discussion period.
>
> As before thank you for your initial review.

---

### Official Review · Reviewer_Vb4W · 2022-10-25

**Confidence:** 4
**Correctness:** 3
**Technical Novelty And Significance:** 2
**Empirical Novelty And Significance:** 2
**Recommendation:** 6

**Clarity, Quality, Novelty And Reproducibility:**

The paper seems to be technically sound and generally meets the quality criteria at major ML conferences. However, the clarity of the paper and the presentation of the content could be significantly improved as outlined in the section above. Further details on the experiment settings would need to be specified for full reproducibility, especially given that corresponding code is missing.
The method itself appears to be novel, but the changes to existing methods (i.e., path integral sampler) seem to be minor and a direct consequence of methods developed for denoising diffusion models. Several additional contributions are interesting from a theoretical viewpoint, however, do not to benefit empirical performance, see also my comments in 'Weaknesses'.

**Strength And Weaknesses:**

**Strenghts:**

The paper shows how to transfer several ideas developed for denoising diffusion models to the task of sampling from unnormalized densities, where the denoising score matching objective cannot be applied. This represents an interesting research direction which is of high interest to practitioners. To this end, the paper derives a suitable objective for learning the drift of the reverse-time process using the Kullback-Leibler divergence on the path-space. Further, the forward process is extended to underdamped Langevin dynamics and the corresponding discrete-time analogous are presented.
Different from previous methods in stochastic optimal control, this leads to a reference process given by an overdamped or underdamped OU process instead of a pinned Brownian motion, which improves the numerical stability of the training procedure (especially for larger number of steps), yields better estimates for the normalizing constants, and allows to construct reverse-time ODEs.

**Weaknesses:**

Comparing the objectives of the proposed method and already existing methods (e.g., the so-called path integral sampler), there is basically only one main difference, i.e., the use of an OU-process instead of a pinned Brownian motion. Judging from the numerical experiments, this can improve training stability and estimation of normalizing constants. It remains unclear whether these benefits stem from the SDE type (variance preserving OU vs. variance exploding Brownian motion), which could also easily be adapted for the path integral sampler, or the initial Gaussian distribution instead of a fixed starting point.

The paper also seems to lack evidence whether these changes systematically improve sample quality (very minor relative improvement in Table 1 and hardly visible differences in Figure 6) and further experiments in this direction would be valuable. Moreover, there are other methods to sample from unnormalized densities using diffusion models, see, e.g., https://arxiv.org/abs/2206.01729 (Section 3.6. and 4.5), which could be compared to the proposed method and which propose to use effective sample size as additional evaluation metric.

While the theory developed in the first sections of the paper is interesting to read, a couple of ideas actually do not benefit the numerical performance in practice, e.g., underdamped Langevin dynamics and the reverse-time ODE (continuous-time normalizing flow), see Appendix C.8 and C.9. I think this might distract the reader from the practically relevant parts and one could, for instance, move corresponding sections to the appendix.

Finally, it would be great if the paper included a discussion why the exponential type integrator is considered in the discrete-time setting. There are better theoretical guarantees as compared to the Euler-Maruyama scheme, however, the latter seems to be used for the standard implementation of the path integral sampler. Also, it would be interesting if the adjoint method could still be employed to compute memory-efficient gradients for the objectives.

**Minor issues:**
1) The proofs of the statements and additional material in the appendix should be referenced in the main paper.
2) 'Logarithmic derivatives of the intractable marginal densities': Should this rather be 'derivatives of the logarithms of the intractable marginal densities'?
3) 'As its initial state $x$ is distributed according to $\pi$': The problem is rather that we cannot sample from the density $\pi$.
4) The time-reversal is written as $(y_t)\_{t \in [0,T]}=(x_{T-t})_{t \in [0,T]}$, which could be interpreted as an equality in an almost sure sense. However, I think that this only holds in distribution.
5) The considered set of path measures should be properly defined.
6) While the chain rule for KL divergences can easily be verified for time-discrete settings, it would be beneficial to include a reference for the time-continuous case in order to see what regularity assumptions are needed.
7) Figure 1 could be described in more detail.
8) I guess the symbol $\perp$ in the beginning of Section 4 should denote the inverse. It feels counter-intuitive to detach the only part of the loss which contains information on the target density and it would be interesting to have more insights regarding this choice.
9) It would be beneficial for the reader to provide more information on the problem settings and metrics.
10) For some settings the presented box plots are difficult to compare and it would help the reader to provide, for instance, relative errors.
11) Typos:
    - 'descent' is missing after 'gradient' on page 2.
    - 'the use *of* underdamped diffusions' on page 2.
    - $\alpha_T$ should probably be $\lambda_T$ on page 2.
    - $\beta_{T-t}$ is missing in the inline equation before (5).
    - The last sentence in Section 2, the sentence starting with 'For example' in Section 3.3, and the second sentence in Appendix C.4 seem to be grammatically incorrect.
    - There is a missing full stop in the last paragraph on page 5.
    - 'At' should be 'As' in the beginning of Section 3.2.
    - Wrong references: 'Figures C.7 and C.7' in Appendix C.7. and 'Table C.7.1' in Section C.7.1.


**Summary Of The Paper:**

The present paper proposes a novel method to approximately sample from unnormalized density functions and estimate their normalizing factors. Motivated by recent successes of denoising diffusion models, the main idea is to reverse an Ornstein-Uhlenberg (OU) process which diffuses the target density into an approximate normal distribution. This requires an approximation to the score of this process which is learned using a neural network by minimizing the reverse Kullback-Leibler distance. Moreover, extensions to underdamped Langevin dynamics and the connection to Schrödinger brides and stochastic optimal control are presented. Finally, the performance of the method is demonstrated by several numerical experiments.

**Summary Of The Review:**

The problem tackled in the paper is of high interest, e.g., in Bayesian statistics and computational sciences. Furthermore, it seems to be a promising direction to transfer methods from denoising diffusion models to improve methods in stochastic optimal control used for sampling from unnormalized densities. The paper provides a good starting point in terms of theoretical contributions, which, however, to a large extend directly follows from the theory developed for denoising diffusion models. The method used for the numerical experiments only differs from existing methods by using a different reference SDE (OU process vs. pinned Brownian motion). This indeed provides improved numerical stability and better estimates of normalizing constants for the considered examples. However, the presentation of the paper could be improved and a full picture of the benefits and drawbacks of the proposed method seems to be lacking.

---

> ### Author Response · Authors · 2022-11-18
> **Response and thank you for your review ! - Part I**
>
> We would like to thank reviewer Vb4W for their in-depth review and careful feedback which have significantly strengthened the manuscript's revised version. We will now proceed to address each of the comments and will reference how the revised version was modified to reflect them.
>
> > " It remains unclear whether these benefits stem from the SDE type (variance preserving OU vs. variance exploding Brownian motion), which could also easily be adapted for the path integral sampler, or the initial Gaussian distribution instead of a fixed starting point …  "
>
> We would first like to highlight that the starting distribution of the path integral sampler cannot be modified to an initial Gaussian as the optimal initial distribution induced by the PIS objective is a point mass. Overall is is not trivial to select a reference SDE such that PIS styled objectives result in a known initial distribution, to the best of our knowledge the OU reference process used in DDS and the pinned Brownian motion used in PIS are one of the very few, however unlike PIS where $X_0=0$ with DDS $X_0$ is approximately distributed as the initial Gaussian as with most DDPM based methods. On the second point indeed the reference process found in the KL divergence objective used by PIS can be changed to an OU process, we have shown that in continuous time such modification coincides with denoising diffusion based approaches, however directly applying the PIS algorithm based on the Euler-Mayurama (EM) discretisation of this modified objective gives very poor empirical performance as can be seen table 5, we can see the method often overestimates ln Z which is very undesirable. This happens as the unconstrained objective in Equation 10 is no longer a proper ELBO when discretized via EM, we have added proposition 2 which formalizes this fact and details the conditions required on $\mathcal{P}^{\mathrm{ref}}$'s discretisation to preserve a valid ELBO. For DDS an exponential type integrator is required to yield a valid ELBO whilst for PIS EM suffices.
>
> > " Moreover, there are other methods to sample from unnormalized densities using diffusion models, see, e.g., https://arxiv.org/abs/2206.01729 (Section 3.6. and 4.5) "
>
> The above work was published at Neurips around 10 days before the ICLR conference deadline and we only became aware of it very close to the submission deadline. That said, we believe the above approach is only suitable for low dimensional settings as it relies on importance sampling to generate the sample paths on which the denoising diffusion is trained on, it is well known that importance sampling does not scale in high dimensions when carried out with respect to an untrained distribution. We can see in this version of the paper that the average dimensionality was of 7.9 ( https://openreview.net/pdf?id=w6fj2r62r_H  Figure 7). Additionally it would be very challenging to carry out a direct comparison as it would require reimplementing their approach without all the specific MD change of basis they use and adapting it to estimate ln Z which would require some additional development of their method. We will acknowledge this approach in the camera ready and provide a conceptual discussion as discussed we dont believe empirical comparison is well suited in this case.

---

> > ### Author Response · Authors · 2022-11-18
> > **Part III - Experimental Results and Differences to Prior Work**
> >
> > > "The paper also seems to lack evidence whether these changes systematically improve sample quality (very minor relative improvement in Table 1 …"
> >
> > We would like to highlight that for larger values of K our approach consistently improves whilst PIS degrades in performance due to many training runs becoming unstable. We can see this across experiments to lead to significant differences (easier to discern in the line plots). Furthermore in Table 1 we can see a non-negligible difference of 0.73 nats in addition to the improvement in Sinkhorn distance ($\mathcal{W}^2_{\gamma=0.01}$). Finally whilst the digit quality itself is not visibly different between PIS and DDS for Figure 6 a careful qualitative inspection shows that we are able to find 9 different digits in the DDS samples whilst we were only able to find 7 for PIS, this is in line with the difference in performance in ln Z and Sinkhorn distance that can be seen in Table 1.
> >
> > Additionally we have added a new section A.2 which analytically computes the drifts for DDS and PIS for a Gaussian target. Corollaries 1,2 show that for a target $\mathcal{N}(\alpha, 1)$ the magnitude for the DDS drift time t=0 scales in $O(\exp(-T))$, whilst for PIS it is $O(1/T)$ showcasing how DDS will have a significantly smaller drift magnitude thus making it more stable.
> >
> > > "For some settings the presented box plots are difficult to compare and it would help the reader to provide, for instance, relative errors."
> >
> > We thank the reviewer for this helpful remark, we have now added line plots which are much easier to read. Furthermore they make clear the competitive advantage DDS has over PIS.
> >
> > > "Further details on the experiment settings would need to be specified for full reproducibility"
> >
> > We have added Tables 7 and 8 which include the optimal hyperparameters for DDS, PIS and Underdamped across all of our experiments. This in combination with the pseudocode and the specification of the target distributions (including the usage of the inference gym package) allow us to fully reproduce our experimental environment.
> >
> > > " but the changes to existing methods (i.e., path integral sampler) seem to be minor and a direct consequence of methods developed for denoising diffusion models."
> >
> > We respectfully disagree with some of these comments, firstly the objective is very different to DDPM and is somewhat more akin to the PIS objective.  Secondly discretizing the objective was done carefully such that the discretized method induces a valid ELBO. Third to the best of our knowledge this is the first paper that has made the connection between stochastic control, inference and DDPM, none of the PIS papers make the connection to DDPM. Finally our approach allows us to obtain a normalizing flow estimate of ln Z which cannot be done by PIS, whilst this did not offer an advantage empirically we suspect it can be leveraged in future work.
> >
> > > "Also, it would be interesting if the adjoint method could still be employed to compute memory-efficient gradients for the objectives."
> >
> > Prior work ([4] Xu, W et al 2021) explores these style of SDE ELBO/Control objectives for the task of Bayesian deep learning and carries out a comparison between classical solvers and the adjoint method (Figure 5 and Appendix C.8) they found both approaches to have the same overall performance and running time (wall clock time). Whilst the adjoint method offered some memory efficiency and has already been adapted to these style of objectives it does not affect the running time and thus we did not find it necessary for our experiments.
> >
> > > "Some of the paper’s claims have minor issues. A few statements are not well-supported, or require small changes to be made correct"
> >
> > Following the very helpful typography and notational corrections from the reviewer we have:
> >
> > * Clarified equalities being in distribution for the time reversal.
> > * Provide referencing for KL chain rule on path measures.
> > * Provided precise statements defining space of path measures.
> >
> > We hope the reviewer can amend this point as we believe that there are no longer any incorrect or unsupported statements.
> >
> > To summarize, we have changed the presentation of our work to clarify some of the major choices and differences required by DDS. In particular our manuscript showcases the need for specialized integrators that preserve P^ref ‘s marginals in order to have a proper evidence lower bound, this is a particularly helpful remark and example for practitioners who wish to develop new samplers and partition function estimators based on these stochastic control objectives.  We have also backed with experimental evidence added to the revised manuscript detailing the need of the specialized integrator. Finally we have provided more clear plots and visualizations which clearly show the stability advantage of DDS over PIS. We hope this addresses the reviewers concerns regarding the clarity and novelty and hope they consider updating their score.

---

> > > ### Author Response · Authors · 2022-11-18
> > > **References for parts I-III**
> > >
> > > [1] Parmas, P., Rasmussen, C.E., Peters, J. and Doya, K., 2018, July. PIPPS: Flexible model-based policy search robust to the curse of chaos. In International Conference on Machine Learning (pp. 4065-4074). PMLR.
> > >
> > > [2] Vicol, P., Metz, L. and Sohl-Dickstein, J., 2021, July. Unbiased gradient estimation in unrolled computation graphs with persistent evolution strategies. In International Conference on Machine Learning (pp. 10553-10563). PMLR.
> > >
> > > [3] Greff, K., Kaufman, R.L., Kabra, R., Watters, N., Burgess, C., Zoran, D., Matthey, L., Botvinick, M. and Lerchner, A., 2019, May. Multi-object representation learning with iterative variational inference. In International Conference on Machine Learning (pp. 2424-2433). PMLR.
> > >
> > > [4] Xu, W., Chen, R.T., Li, X. and Duvenaud, D., 2022, May. Infinitely deep bayesian neural networks with stochastic differential equations. In International Conference on Artificial Intelligence and Statistics (pp. 721-738). PMLR.

---

> > ### Author Response · Authors · 2022-11-18
> > **Part II - Exponential Integrator and Detach Gradient**
> >
> > > "Finally, it would be great if the paper included a discussion why the exponential type integrator is considered in the discrete-time setting. There are better theoretical guarantees as compared to the Euler-Maruyama scheme, however, the latter seems to be used for the standard implementation of the path integral sampler. ..."
> >
> > Without the exponential integrator we are no longer guaranteed to have a valid ELBO and thus the approach can overestimate ln Z, leading to poor performance as can be seen in Table 5. We now provide a formal argument that given a reference SDE with path measure $\mathcal{P}^{\mathrm{ref}}$, then the integrator for $\mathcal{P}^{\mathrm{ref}}$must exactly preserve the time marginals of $\mathcal{P}^{\mathrm{ref}}$, otherwise we are no longer guaranteed an ELBO. This has now been added in Proposition 2 of section 3.1 . We believe this is a useful contribution and partly where our work stands out as it informs the careful design choices that are required when adapting stochastic control based methods to partition function estimation and sampling. To conclude DDS lacks estimation guarantees when using the EM integrator and thus an exponential type integrator is strictly required.
> >
> > |         |         Funnel         |                    |              |       Ionosphere       |                      |              |
> > |---------|:----------------------:|--------------------|:------------:|:----------------------:|:--------------------:|:------------:|
> > |         | Exponential Integrator | Euler Mayurama     | Ground Truth | Exponential Integrator |    Euler Mayurama    | Ground Truth |
> > | $k=64$  | $-0.206 \pm 0.059$     | $2.425 \pm 0.266$  |      $0$     |  $-111.693 \pm 0.169$  | $-106.364 \pm 0.371$ |  $-111.560$  |
> > | $k=128$ | $-0.176\pm0.099$       | $2.011\pm 0.532$   |              |  $-111.587 \pm 0.136$  |  $-111.158\pm 0.399$ |              |
> > | $k=256$ | $-0.221 \pm 0.076$     | $-0.086 \pm 0.124$ |              |   $-111.575\pm 0.163$  | $-112.271 \pm 0.366$ |              |
> > | $k=512$ | $-0.176 \pm 0.068$     | $-0.154 \pm 0.066$ |              |   $-111.582\pm 0.136$  | $-112.912 \pm 0.353$ |              |
> >
> > We include the above ablation in the revised version of the manuscript which showcases how the EM based integrator approach overestimates ln Z  for k=64,128 and overall underperforms compared to the exponential integrator based DDS.
> >
> > > "It feels counter-intuitive to detach the only part of the loss which contains information on the target density and it would be interesting to have more insights regarding this choice"
> >
> > In addition to the provided empirical evidence our motivation behind this choice arises from prior work as it is known that optimization through unrolled computation graphs can be chaotic and introduce exploding/vanishing gradients ([1] Parmas et al., 2018; [2] Metz et al., 2020) which can numerically introduce bias. Detaching gradients of log densities in recognition networks has been successfully applied in related prior work ([3] Greff et al., 2019) where detached scores are provided as features to the recognition networks. We have added this discussion to Appendix C.8.1.  The symbol ⊥ was meant to denote detach following similar nomenclature in previous VI papers.
> > Finally this is not the only part of the loss that contains information on the target density.  The loss itself (see the last term of Eq 10) has a terminal term which involves the target density, this is not detached and is back propagated through, we do not detach any terms in the loss we simply detach the gradient term in the neural network drift parameterization which helps stabilizing the dynamics at training time.

---

> > ### Comment · Reviewer_Vb4W · 2022-11-28
> > **Thank you for the response and revision!**
> >
> > I thank the authors for their response, the revised version, including new theoretical results, and the additional experiments. This definitely adds significant value to the paper and clarifies several of my issues.
> >
> >
> > I have a couple of follow-up questions:
> > 1. Thank you for the explanation regarding detaching of the score. Did I understand it correctly, that the implementation uses the parametrization $\mathrm{NN}_1(k,x;\theta) + \mathrm{NN}_2(k,\theta) \circ \left( \nabla \pi (x) \right)^\perp$. I assume that the $\circ$ symbol refers to multiplication (as in the PIS paper) and perhaps this should be replaced by $\cdot$ to prevent confusion with a composition.
> > 2. I am still not sure, if it is possible to fully reproduce the experiments from the paper. For instance, the PIS implementation uses gradient and output clipping, weight decay, and learning rate scheduling. It is unclear how these hyperparameters are handled in the present paper. Moreover, there seems to be no information on how the "gold standard" for $Z$ is computed and why this is a realiable estimate (except for examples such as the Funnel distribution where $Z$ is known).
> > 3. I think that the paper still does not mention early enough that extensions such as underdamped Langevin dynamics as well as the continuous-time normalizing flow did not lead to improvements in practice.
> >
> >
> > Finally, I would be glad if you could answer the following further questions:
> > 1. The proof of Propostion 1 claims that the Radon-Nikodym derivative follows directly from Girsanov's theorem.
> > However, only the terminal conditions of $\mathcal{Q}^\theta$ and $\mathcal{P}^{\mathrm{ref}}$ are the same (the change of initial condition leads to the terminal cost in the objective) and it seems that Girsanov's theorem is actually applied to the corresponding reverse time-processes of $\mathcal{Q}^\theta$ and $\mathcal{P}^{\mathrm{ref}}$?
> >
> >     Similarly, this is handled in equation (7). I think this should be clarified. Moreover, should the order of $p_T$ and $\mathcal{N}(0,\sigma^2I)$ be reversed in the KL divergence?
> > 2. It seems that the KL-divergence in (7) can also be computed without the reparametrization which leads to (10). Some other works, see [here](https://link.springer.com/article/10.1007/BF01448198) and [here](https://arxiv.org/abs/2211.01364), compute this KL-divergence via the associated HJB equation. While you correctly state that the objective in (7) cannot be tackled directly, the latter reformulations seem to yield a viable alternative. Did you also consider this route?

---

> > > ### Author Response · Authors · 2022-11-28
> > > **Thank you for your questions and further corrections ! - Part II**
> > >
> > > > 1. The proof of Proposition 1 claims that the Radon-Nikodym derivative follows directly from Girsanov's theorem. However ...
> > >
> > > We indeed apply Girsanov theorem to the time reversals, however note that the RND between two path measures and their respective time reversals are the same. However you are indeed right that we should mention this intermediate step and detail in the proof specifically how Girsanov’s theorem is applied to the time reversals. We will include this in the final version of the manuscript.
> > >
> > > > Moreover, should the order of  and  be reversed in the KL divergence?
> > >
> > > Yes this is indeed a typo. We have amended this in our version and will update for the final version of the manuscript.
> > >
> > > > 2.  It seems that the KL-divergence in (7) can also be computed without the reparametrization which leads to (10). Some other works, see here [6] and here [5], ...
> > >
> > > We were neither aware of [6] nor [5] at the time of writing; [5] appeared about a month after our paper submission. However, we did indeed explore a divergence based formulation of the KL (i.e. involving $\nabla \cdot f$) as we were able that such a formulation is possible; e.g.  both Proposition 1 of  [3] or Proposition 9 of [4] show that scores can indeed be traded for divergences of drifts. Whilst these reformulations are indeed very interesting, we believe that they are computationally less suitable than the expressions we propose (note that theoretically they are equivalent):
> > >
> > > 1. Computing the divergence of the network drift is expensive, our approach does not require estimating divergences. It is true that the divergence can be estimated in an unbiased fashion using Hutchinsons trace estimator. However this introduces an additional source of error into the loss, and still requires some additional computation at train time.
> > > 2. With the divergence based expressions it is no longer clear how to preserve a valid ELBO in discrete time as the HJB (or FPK based as in [3,4]) results used to trade of the score with the divergence only hold in continuous time. A lot more work would be required in order to adapt this such that we have an estimator of the normalizing constant whose logarithm has an expectation lower bounding the true normalizing constant. We suspect more results must be developed in order to apply formulations such as in [5] to partition function estimation as we do.
> > >
> > >
> > > [1] Zhang, Q. and Chen, Y., 2021. Path Integral Sampler: a stochastic control approach for sampling. arXiv preprint arXiv:2111.15141.
> > >
> > > [2] Arbel, M., Matthews, A. and Doucet, A., 2021, July. Annealed flow transport monte carlo. In International Conference on Machine Learning (pp. 318-330). PMLR.
> > >
> > > [3] Vargas, F., 2021. Machine-learning approaches for the empirical Schrödinger bridge problem (No. UCAM-CL-TR-958). University of Cambridge, Computer Laboratory.
> > >
> > > [4] Liu, G.H., Chen, T., So, O. and Theodorou, E.A., 2022. Deep Generalized Schr\" odinger Bridge. arXiv preprint arXiv:2209.09893.
> > >
> > > [5] Berner, J., Richter, L. and Ullrich, K., 2022. An optimal control perspective on diffusion-based generative modeling. arXiv preprint arXiv:2211.01364.
> > >
> > > [6]  Pavon, M., 1989. Stochastic control and nonequilibrium thermodynamical systems. Applied Mathematics and Optimization, 19(1), pp.187-202.

---

> > > ### Author Response · Authors · 2022-11-28
> > > **Thank you for your questions and further corrections ! - Part I**
> > >
> > > > Thank you for the explanation regarding detaching of the score. Did I understand it correctly, that the implementation uses the parametrization … . I assume that the  symbol refers to multiplication (as in the PIS paper) ...
> > >
> > > Your interpretation is indeed correct. The symbol represents element wise multiplication and we will replace it with $\cdot$ as you have suggested in the camera ready version of the paper. Thank you for this pointer.
> > >
> > > > I am still not sure if it is possible to fully reproduce the experiments from the paper. For instance, the PIS implementation uses gradient and output clipping, weight decay, and learning rate ...
> > >
> > > We followed the PIS repos clips, and used the same base settings for clipping they had (from what we were able to infer from the repo as most config files are missing). That is :
> > >
> > > 1. We clip gradients (in the PISgrad net) with 1.0e02
> > > 2. We clip the full network drift with 1.0e04
> > >
> > > We will add these details (final version) to the main paragraph of Appendix C.
> > >
> > > Regarding optimizer settings (e.g. weight decay, lr decay, ...) we use the default Adam optimiser settings found in the Jax Optax module. We report the learning rate used but other than this we did not change any other parameters. We will add a further table with learning rate and the default opt settings we used in the appendix.
> > >
> > > Furthermore we will be more clear about the NN frameworks we use (e.g. jax + Haiku and default initialisation for intermediate weights).
> > >
> > > Finally we aim to open source the code. Again we thank the reviewer, we believe that with these extra settings we should be able to closely reproduce the results. To the best of our knowledge other than seeds, there are no other free parameters.
> > >
> > > > Moreover, there seems to be no information on how the "gold standard" for …  is computed ...
> > >
> > > The gold standard as with prior work was computed with a long run tuned SMC chain (1000 temperatures, 30 seeds with 2000 samples each) . We briefly mention this in the Euler vs Exponential integrators section (but did not give the number of temperatures). We should have stated this in the first paragraph of the experimental section. This was accidentally commented out when trying to comply with the 9 page limit. Thank you for highlighting this very important detail.
> > >
> > > Note this is a reliable estimate and has been used in prior works such as [1,2] in particular for all the unimodal Bayesian examples considered. For our main multimodal example (NICE) we know exactly the normalizing constant as with the Funnel distribution.  We will clarify these important details. Thank you for highlighting these. We will include all these changes in the experimental details sections of our manuscript.
> > >
> > > > I think that the paper still does not mention early enough that extensions such as underdamped Langevin dynamics as ...
> > >
> > > It is indeed correct that these proposed extensions have not provided improvements in practice. We believe it is worth detailing them as we conjecture that better integrators, training techniques or better parameterization of the value functions for underdamped could lead eventually to significant gains. We will mention this (i.e. not incurring in performance gains) both in the beginning of the experimental section as well as the first instance these approaches are mentioned, which is the last paragraph of the introduction. These are changes we can make seamlessly within the page limit.

---

> ### Author Response · Authors · 2022-12-06
> **Reminder - End of phase 2 dicussion period soon (12 of December)**
>
> Dear Reviewer,
>
> This is a short reminder that the phase 2 dicussion period is coming to an end on the 12th of December.  As mentioned by the reviewer the revised version has clarified/addressed most issues as well as "added significant value to the paper", we hope that given these significant improvements and the latest clarifications we provided the reviewer may consider updating their main review/score. Furthermore we are happy to address any outstanding questions in the remainder of this period.
>
> As before we thank the reviewer for their review and the prompt follow up questions.

---

> > ### Comment · Reviewer_Vb4W · 2022-12-08
> > **Updated recommendation**
> >
> > Thank you for the clarifications and for outlining the updates in the camera-ready version. I have updated my recommendation accordingly.
> >
> > Some comment on the "ground truth" log-normalizing constants (see https://openreview.net/forum?id=8pvnfTAbu1f&noteId=I8w-iW1EuK): I know that also in prior work the gold standard was computed with a long run tuned SMC chain. However, it depends on the problem whether this yields a "reliable estimate", and I think it is quite important to discuss this in the paper (given that this is the ground-truth we are comparing to). Perhaps, one could report lower and upper bounds using AIS or add additional experiments with explicitly computable normalizing constants?

---

> > > ### Author Response · Authors · 2022-12-09
> > > **Thank you for the updated recommendation and the further feedback.**
> > >
> > > We would like to thank the reviewer for reassessing our work and continuing to provide excellent feedback. For the final version we will add the following:
> > >
> > > 1. A discussion of the gold standard estimate in the context of each distribution.
> > > 2. Additional results with the difficult mixture target which is also explored in the PIS paper and has a known ground truth. This in conjunction to the Funnel and NICE tasks bring our examples with known ground truths to 3. We can also add the student-T target from [1].
> > >
> > > [1] Doucet, A., Grathwohl, W., Matthews, A.G. and Strathmann, H., 2022. Score-based diffusion meets annealed importance sampling. arXiv preprint arXiv:2208.07698.

---

### Author Response · Authors · 2022-11-18
**Main Response**

We would like to thank all reviewers for taking the time to review this manuscript and for providing insightful feedback and discussion which has itself strengthened the manuscript overall. We have now uploaded the revised version of the manuscript which we hope reflects most of the advice provided by the reviewers.

In this general response we would like to highlight the overall changes made to the presentation of the paper and list the new results we have added.

1.  We added a new section (3.1) detailing the need for the exponential integrator when discretizing DDS. In proposition 2 we formally state and prove the conditions required when discretizing sampling approaches based on control objectives such that the objective itself and expected ln Z estimators are valid ELBOs. This is an important contribution and remark as it details the careful design required when designing integrators for these family of approaches when aplied to inference. Note this is not required for generative modeling and it illustrates the challenges that arise when porting over DDPM based approaches from generative modeling to sampling.
2. We have added Corollaries 1, and 2 to Appendix A.2, which show the analytical optimal PIS and DDS drifts for the gaussian target setting. This allows us to compare the magnitudes of both drifts motivating why DDS is more numerically stable.
3.  We provided experimental evidence comparing the exponential and Euler based integrators in Table 5. As the theory we develop motivates the Euler based discretisation overestimates the partition function in practice.
4. We have added Algorithm 1 detailing the pseudocode for DDS to the main section of the paper. This should make the work more accessible to practitioners.
5. We replaced the boxplots with line plots that are much more clear and easier to read. The new plots allow to visualize the differences with the approaches more clearly addressing the comments pertaining to the readability of the box plots.
6. We moved the discretisation of the underdamped dynamics to the appendix in order to focus the reader and improve the overall presentation of the paper.
7. We have added tables 8 and 9 detailing all the optimal hyperparameters for DDS and PIS across all experiments. This in conjunction with algorithm 1 will allow a reader to reproduce our setup as all the target distributions are completely specified.
8. We highlight our empirical comparisons to Langevin Monte Carlo as well as other MCMC methods (UHA, MCD, LDVI). These results can be found in Tables 3, and 4 of the Appendix.
9.  At the time of submission we were unaware we could submit a main file a pdf that included the appendix we have done this for the revised version and we hope it facilitates reading.
10. Finally we would like to remark that we carry out a total of 7 target-distribution experiments (funnel, lgcp, ion, sonar, brownian, nice, vae) , with an average a dimensionality of 280.14 (max dim being 1600) for the target distribution. Within Monte Carlo sampling this is considered high dimensional. Furthermore we have performed several ablations of our proposed features (e.g. stop gradient, exp integrator, drift magnitude, probability flow ODE) thus we believe the overall scale of our evaluations to be suitable. Typically most sampling papers explore 4-5 targets out of the 7 we have presented here, notice the NICE target was only proposed a few months ago.

We hope that with the new presentation, and the added results (both thereotical and experimental) we have addressed most of the reviewers concerns, as before we thank the reviewers for their time and efforts in evaluating this work.

---

### Author Response · Authors · 2022-12-12
**Final day of discussion period Phase II (today) - No acknowledgement of revised version / rebuttal from most reviewers.**

Dear Reviewers and Area Chairs,

Today is the final day to provide feedback/responses to our rebuttal / revised version of the manuscript. With the exception of reviewer Vb4W we have not received any acknowledgement of our revised version / rebuttal and would really appreciate feedback regarding the accessibility of the new draft, as well as whether the new results/experiments we provided in conjunction with the rebuttal address the reviewers' concerns.

We spent a significant amount of time in the revised version, from new conceptual results to new experiments with the hope of addressing the reviewers concerns, additionally we restructured the presentation (including adding more legible figures) of the manuscript in the hope to improve readability aimed at addressing reviewers **cpvn** and **bg9B** comments. It would be extremely helpful to obtain feedback / updated evaluations regarding whether many of these concerns were addressed. Our understanding is that the rebuttal and discussion thereof are core components to ICLRs review process.

As before we thank all the reviewers for their initial response and reviewer Vb4W for the updated review and thorough evaluations of our rebuttal/revised manuscript.

**Update:** We would like to thank reviewers Vb4W, SEx8, and bg9B for their prompt responses and updates. Unfortunately we did not recieve a response/acknowledgement from reviewer **cpvn**.

Regards,

Authors of Paper 5353

---

### Decision · Program_Chairs · 2023-01-20

**Decision:**

Accept: poster

**Justification For Why Not Higher Score:**

The paper could be presented as a spotlight given the practical and theoretical contributions and given that the experimental evaluation was beefed up during the rebuttal phase.

**Justification For Why Not Lower Score:**

Authors addressed the concerns raised by the reviewers and made substantial improvement to the submitted paper.

**Metareview: Summary, Strengths And Weaknesses:**

This work takes inspiration from ideas developed for denoising diffusion models to sample approximately from unnormalized densities. Reviewers acknowledged contributions of this work, which paves the way for a new interesting research direction. Reviewers pointed to weaker experimental results in the submission, but the authors provided the additional data required to convince reviewers of the practical relevance of the contributions.

**Note From Pc:**

if the above contains the word "oral" or "spotlight" please see: "oral" presentation means -> notable-top-5% and "spotlight" means -> notable-top-25%. As stated in our emails, we are disassociating presentation type from AC recommendations

**Summary Of Ac-Reviewer Meeting:**

N/A